# Reassessing chain tilt in the lamellar crystals of polyethylene

Shusuke Kanomi [1,2], Hironori Marubayashi [3], Tomohiro Miyata[3] & Hiroshi Jinnai [3] ✉

Semicrystalline polymers are extensively used in various forms, including fibres, films, and bottles. They exhibit remarkable properties, e.g., mechanical and thermal, that are governed by hierarchical structures comprising 10–20-nm-thick lamellar crystals. In 1957, Keller deduced that long polyethylene (PE) chains fold to form thin single lamellar crystals, with the molecular chains perpendicular to the flat faces of the crystals (the chain-folding model). Chains inclining to the perpendicular orientation in single crystals have since been reported, along with their effects on the physical properties of PE. For bulk specimens, the chain tilt angle ($\varphi$) has been investigated only for model samples with well-annealed internal structures. However, for briefly annealed specimens, the $\varphi$ values of lamellae and their origins are controversial owing to the disordered lamellar morphology and orientation. Herein, we report the direct determination of molecular-chain orientations in the lamellar crystals of high-density PE using a state-of-the-art electron-diffraction-based imaging technique with nanometre-scale positional resolution and provide compelling evidence for the existence of lamellar crystals with different inner-chain orientations. Clarifying the nanoscale variation in lamellar crystals in PE can allow precise tuning of properties and expedite resource-saving material design.

Semicrystalline polymers, such as polyethylene (PE), are used to prepare fibres, films, and bottles because of their processability and thermal stability. Polymer crystallisation can induce unique features owing to the specific molecular shape of long chains. Flexible polymer chains fold regularly to form 10–20-nm-thick lamellar crystals (folded-chain crystals; FCCs)[1,2], which grow with 10–20-nm-thick amorphous regions to form higher-order structures, such as spherulites (FCC-containing spherical assemblies; diameter = 1–1000 μm)[2,3]; typical spherulite-containing PE specimens exhibit a tensile modulus of 0.1–1 GPa[4,5]. Alternatively, flow-induced crystallisation yields 'shish kebab' structures comprising nearly fully extended-chain crystals (ECCs) and FCCs as 10–100-μm-long 'shish' and 'kebabs,' respectively[2,6]. ECCs have an excellent tensile modulus (220 GPa)[7] comparable to that of an infinitely thick theoretical perfect crystal[8]. Thus, the crystallisation-dependent hierarchical structures (Supplementary Fig. 1) significantly affect the physical properties of the polymers. The fraction, size, and orientation of the hierarchical structural elements, including lamellar crystals, higher-order structures[9], chain conformations, and unit-cell packing[10], significantly affect the optical, thermal, and mechanical properties.

The crystal structures of semicrystalline polymers (such as unit-cell packing and chain conformation) have attracted considerable attention since the 1930s. The PE unit-cell containing chains parallel to the crystallographic $c$-axis is shown in Fig. 1a[11]. In 1957, Keller first established the chain-folding model by relating the transmission electron microscopy (TEM) images and electron diffraction (ED)

[1]Department of Applied Chemistry, School of Engineering, Tohoku University, 6-6 Aramaki Aza Aoba, Aoba-ku, Sendai, Miyagi 980-8579, Japan. [2]Science & Innovation Center, Mitsubishi Chemical Corporation, 1000 Kamoshida-cho, Aoba-ku, Yokohama, Kanagawa 227-8502, Japan. [3]Institute of Multidisciplinary Research for Advanced Materials, Tohoku University, 2-1-1 Katahira, Aoba-ku, Sendai, Miyagi 980-8577, Japan. ✉e-mail: hiroshi.jinnai.d4@tohoku.ac.jp

patterns of PE single lamellar crystals, and the chain orientation (*c*-axis) was assumed to be perpendicular to the flat faces of the lamellar crystals ({001} facet) (Fig. 1b and Supplementary Fig. 1b)[1]. The chain-folding concept was deduced from the ordered-folding model, in which tightly folded chains re-enter the lamellar crystal in the adjacent position (Fig. 1d)[2,12]. However, the flat-face-adjacent amorphous region is denser than the inner part of the lamellar crystals, resulting in a density anomaly (Supplementary Fig. 1b)[13,14].

The following factors have been suggested to affect the degree of the density anomaly (Fig. 1e): (i) the chain tilt in the crystallite, (ii) the fraction of the tightly folded chains (sharp folds) on the flat faces of lamellae, and (iii) the fraction of the chain ends on the flat faces[14]. These factors affect the polymer chain overcrowding and consequently the total free energy. Increasing the chain tilt angle $\varphi$ (Figs. 1b, c) reduces the area density of folded chains because the flat-face area per chain increases by a factor of $1/\cos\varphi$[13–15]. A moderate fraction of sharp folds (~1/3) mitigates the chain density near the flat faces[14]. Chain ends on the flat faces make space for adjacent folded

chains[14]. Although these three factors work cooperatively to mitigate the density anomaly, the relationships between them have not been fully evaluated, owing to the difficulty of observing the molecular-scale structures in real space[14]. Because the chain tilt in lamellar crystals has a larger structural scale than that of the other two elements (chain folds and ends on the flat faces of lamellar crystals), nanoscale observation of chain tilt is a key to understanding the relationships between the aforementioned factors. Furthermore, chain tilt affects the lamellar surface energy[13] and related properties (such as the melting point)[16–18]. For instance, $\varphi$-dependent differences have been observed in the melting points of single crystals[18]. Thus, the experimental investigation of the chain tilt is important from the viewpoint of structure–property relationships.

Around 1960, Bassett et al. reported that chains in single crystals tilt from the perpendicular orientation by $\varphi$ (Supplementary Figs. 2a, b)[19–21]. Around 1980, chain tilt was also reported for bulk specimens (Supplementary Figs. 2c, d)[22–25]. From 1978 to 1981, Bassett and Hodge reported ~30° as the most commonly observed $\varphi$ value for well-annealed high-

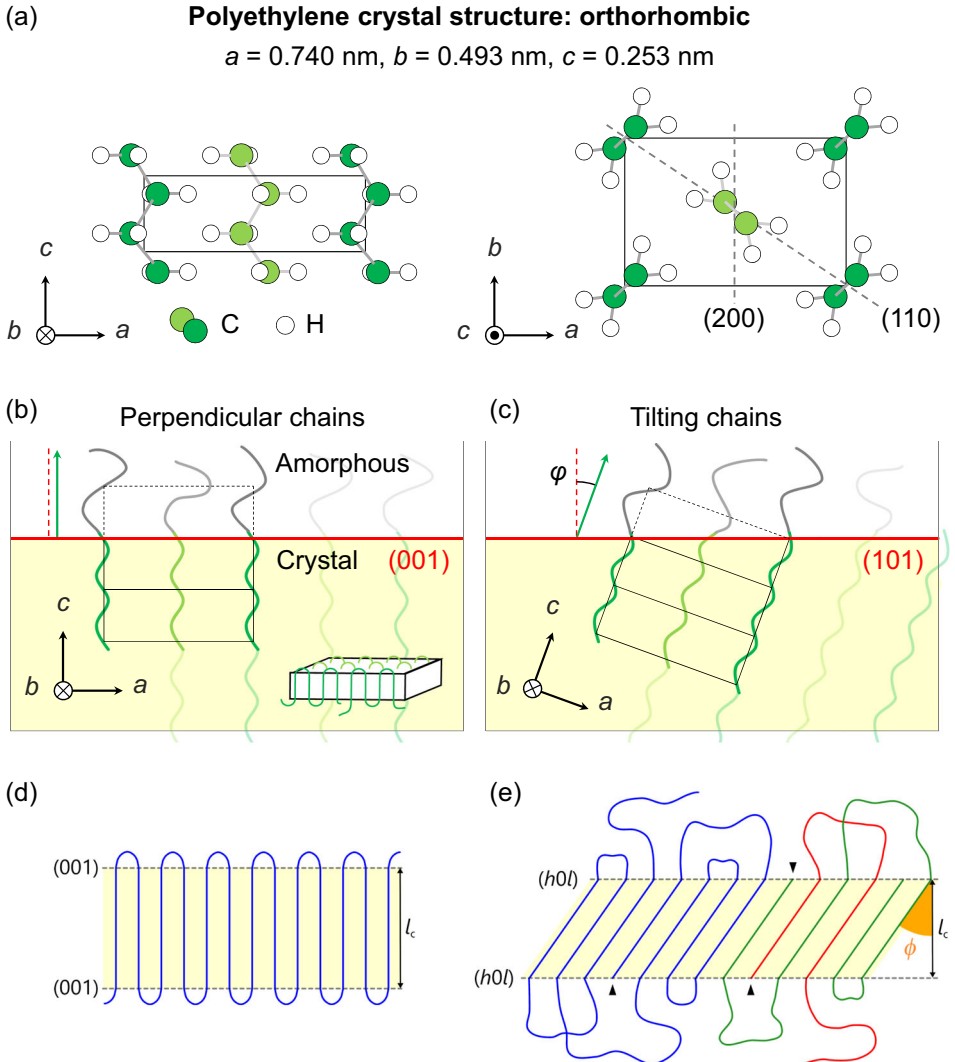

**Fig. 1 | Schematic of PE lamellar structures. a** Crystal structure of orthorhombic polyethylene (PE)[11]. Two types of planar-zigzag-structured PE chains—shown as a light-green central chain and dark-green corner ones—occupy the unit cell with different setting angles (the angle between the *bc* and zigzag planes).
**b, c** Schematics of the molecular-chain orientation in a lamellar crystal with a **b**, perpendicular and **c**, non-perpendicular (tilting) *c*-axis. The crystalline region is shown in yellow, and the flat faces of the lamellar crystals are highlighted in red. The

angle between the red dashed line (normal to the flat face of the crystal) and green arrow (molecular chain axis; *c*-axis) is the chain tilt angle $\varphi$. The flat-face area increases by a factor of $1/\cos\varphi$. **d** Ordered-folding model featuring adjacent re-entry of regular sharp folds. **e** Density-anomaly-free modified-folding model[14], in which the crystalline region, flat faces, lamellar thickness ($l_c$), and chain ends are represented by a yellow area, dotted lines, a black arrow, and triangles, respectively. Diverse chains are coloured differently.

molecular-weight PE specimens, regardless of the molecular weight and crystallisation temperature ($T_c$)[22,23]. Thermodynamics indicates that the chain tilt-created facets ($\varphi \neq 0°$) reduce the chain density near the flat face and the steric repulsion between adjacent folds, increasing the entropy and thus reducing the free energy[15]. Additionally, molecular simulations have corroborated chain tilt. In 2000, Gautam et al. found that chain-tilt-induced {$h0\ell$} facets are more favourable than the {001} facet (Fig. 1b) by analysing the density profiles and interfacial energies at the crystalline/amorphous interfaces[13]. The calculated interfacial energy of PE was minimised at a certain $\varphi$ value (34°, {201})[13]; these data are used in current molecular-simulation studies[14,26].

However, in 1981, Voigt-Martin and Mandelkern demonstrated the coexistence of various lamellar crystals ($\varphi = 18°$, 34°, and 46°) in bulk PE specimens with low molecular weights. The $\varphi$ differed depending on $T_c$ (that is, the thermal history)[24]. The chain tilt was 'indirectly' estimated from the morphological features of lamellar crystals by assuming the chain orientation (Supplementary Figs. 2c, d). Furthermore, because the molecular-chain tilt in the lamellar crystals of bulk PE specimens that are briefly annealed has never been visualised in real space (the internal structures are not adequately equilibrated, and the lamellar crystals are in a growing stage), the variation in chain tilt and the formation mechanisms of lamellar crystals at each location in these suboptimal PE specimens are controversial.

Various experimental techniques have been employed to investigate polymer crystals. Conventional X-ray diffraction[11,27], spectroscopy[28,29], and optical microscopy[3] only provide spatially averaged, micrometre-to-millimetre-scale structural information of the hierarchical crystalline structure. Although atomic force microscopy can visualise manifold hierarchical morphologies (from chain folding[30] to spherulites[31]), it yields surface-exclusive information. TEM can reveal the lamellar morphology in bulk specimens through contrast enhancement (electron staining)[32,33]. However, the inner-chain-related information is obfuscated by chemical reactions between the molecular chain and staining agent. The conventional-TEM-derived structural information on lamellar crystals is also limited.

Recently, a new scanning TEM (STEM)-based method[34–36] called nanodiffraction imaging (NDI) was used to directly analyse crystal orientations in unstained high-density PE (HDPE)[37]. In this method, an electron nanobeam is raster-scanned across the specimen to acquire two-dimensional (2D) ED patterns for local structural analysis with significantly higher resolution than conventional methods (Fig. 2a)[38]. The recently developed high-speed, high-sensitivity direct electron detectors (DEDs)[39] have broadened the applicability of low-dose NDI for electron-irradiation-sensitive materials, including polymeric systems, without causing severe electron irradiation damage[37,40–44]. Because it permits image reconstruction from diffraction data, NDI can provide direct structural information on semicrystalline polymers without electron staining.

In this study, the nanoscale morphologies of lamellar crystals and the molecular-chain orientations in lamellae were visualised simultaneously in a 'direct' manner using position-resolved ED patterns. Chain tilt in lamellae was experimentally identified without any assumptions through orientational relationships between the lamellae and their inner-chain orientations. Precise analysis revealed the variations in the lamellar morphology and the chain tilt inside lamellae for an HDPE specimen prepared via short-duration annealing.

## Results and Discussion

First, a pressed HDPE film melt-crystallised under thermal annealing at 120 °C for 60 min was characterised using conventional wide-angle X-ray diffraction (WAXD), small-angle X-ray scattering (SAXS), and staining-TEM. The WAXD and SAXS analyses indicated that the specimen had a stacked lamellar structure (crystallinity = ~60%, non-oriented) with 18-nm-thick lamellae ($l_c$) and a 13-nm-thick amorphous layer ($l_a$) (Supplementary Fig. 3; see Supplementary Information). Certain ultrathin (~100 nm thick) sections were stained with $RuO_4$ for TEM analysis (Supplementary Fig. 4), which revealed stacked lamellar structures with an undulating stripe morphology. Because the lengths of most lamellar crystals in their in-plane direction exceeded the thickness of the ultrathin section, the edge-on lamellar crystals were considered to penetrate the section. However, staining methods often result in over/understaining; thus, the observed thicknesses of the lamellar crystals in stained specimens are inaccurate[27].

To initiate NDI, which was performed on unstained sections, a 1.2-nm-diameter electron beam was scanned across the specimen via STEM with a step (6 nm) smaller than both $l_c$ and $l_a$. The ED patterns at each point were recorded using a DED, and 360,000 ED patterns were acquired from a 3.6-$\mu m^2$ area. No distinct morphologies were visible in the bright-field (BF) STEM image acquired after NDI, because the specimen was unstained (Fig. 2b; white dashed square: area scanned for NDI). Moreover, the number, intensity, and azimuth of the ED peaks varied with respect to the scanning position (Fig. 2c–f). The ED peaks were assigned to the 110, 200, 020, and 220 reflections of orthorhombic PE[11]. A convergent electron beam produced disc-like peaks with a radius corresponding to a convergence semi-angle $\alpha$ (Fig. 2a). The averaged ED pattern of the entire area appeared as a ring (Fig. 2g), similar to that obtained using a 2.1-$\mu m$-diameter parallel beam (Fig. 2h), indicating a random crystal orientation within the target volume.

The chain axis in orthorhombic PE is parallel to the $c$- and $c^*$-axes[11]. Therefore, the two-spot $hk0$ peaks symmetric about the beam centre (Figs. 2c, d) indicated that the chain orientation was perpendicular to the line connecting the two peaks. Moreover, the hexagon-like $hk0$ pattern (Fig. 2e) represents the incident electron beam parallel to the chain axis (incident to [001]). However, for polycrystalline regions (Fig. 2f), the relationship between the orientations of the incident electron beam and molecular chain was not easily determined. These results indicate that NDI can detect crystal information in the nanoscale regions of semicrystalline polymers.

Subsequently, dark-field (DF) STEM images were reconstructed using the two highest-intensity centre-symmetric 200-reflection peaks in the image-processed ED patterns (Supplementary Fig. 5 and Supplementary Discussion). A stripe morphology was clearly visible in the reconstructed DF-STEM image of the *unstained* specimen (Fig. 3a), similar to that of the TEM-analysed *stained* specimen (Supplementary Fig. 4). The width of the bright stripes (17.0 ± 2.8 nm) and the distance between neighbouring bright stripes (28.3 ± 4.1 nm) were consistent with $l_c$ (18 nm) and the long period ($L_p = l_c + l_a = 31$ nm), respectively. Therefore, these bright stripes were undoubtedly lamellar crystals. Because the contrast of the reconstructed image is directly generated from diffraction intensities, NDI can identify lamellar crystals far more accurately than conventional methods. In contrast, STEM images were created virtually using the intensities of the circular BF and annular DF (ADF) areas in each ED pattern (Supplementary Fig. 6). The NDI-derived 'virtual' BF and ADF-STEM images showed indistinct contrast, highlighting the difficulty of visualising lamellar crystals even with a carefully tuned conventional STEM technique[34–36].

In the reconstructed DF-STEM image (Fig. 3a), nine regions that included stacked lamellae (indicated by red boxes) were selected to analyse the chain tilt in lamellar crystals. The chain orientation angles in the Cartesian coordinate system (absolute molecular-chain orientations) were plotted (Fig. 3b). Because the lamellar crystals shown in Fig. 3a were reconstructed using the intensities of the 200 spots and viewed edge-on (with most lamellae having thicknesses similar to that obtained via SAXS, that is, ~18 nm), the chains in the lamellae were oriented along the in-plane direction of the image (see Supplementary Information and Supplementary Figs. 7 and 8 for details). Because the growth direction of PE lamellae is parallel to the $b$-axis, the edge-on lamellae grow perpendicular to the in-plane direction of the image.

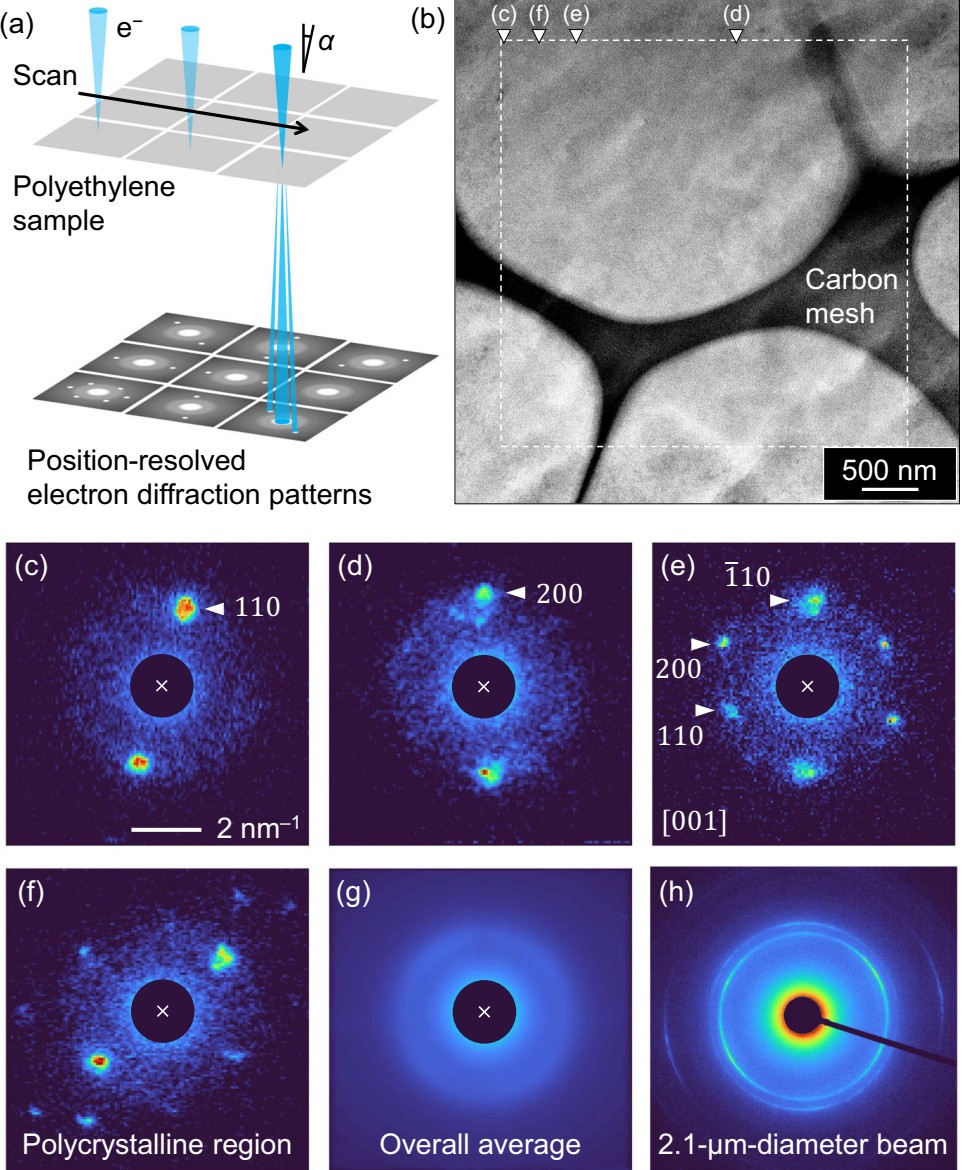

**Fig. 2 | Schematic of nanodiffraction imaging. a** Schematic describing nanodiffraction imaging (NDI). **b** Bright-field scanning transmission electron microscopy (STEM) image of high-density PE (HDPE) acquired after NDI. The dashed square indicates the NDI scan area. **c–e** Indexing representative electron diffraction (ED) patterns acquired via the NDI of HDPE, which were extracted from the tips of the white triangles shown in (**b**). In (**e**), the direction of the incident electron beam is indicated by [*uvw*], where *u*, *v*, and *w* are the smallest integers with no common divisor. **f** ED pattern of a polycrystalline region. **g** Average of all the ED patterns. **h** ED pattern of the same sample acquired from another field of view using a 2.1-μm-diameter electron beam.

First, the morphology and inner-chain orientation of individual lamellar crystals are worth discussing. While Fig. 3c shows a straight lamella with a uniform chain orientation, Fig. 3d shows a long, curved lamella with a uniform chain orientation; the latter probably grew in out-of-plane directions of the lamella while maintaining the chain orientation.

Apart from the lamellae with uniform chain orientations, certain lamellae exhibited multiple orientations internally. Figure 3e shows a straight lamella with multiple chain orientations (with a deviation of ~34°). Two formation mechanisms may apply to this lamella: (i) collision of several separately nucleated crystallites and (ii) noncooperative changes in the orientations of the chains in the lamella during annealing, with the chains in each local region of the lamella reorienting in different directions[45]. In Fig. 3f, the upper and lower parts of the lamella exhibit chain orientations that differ by >20°, indicating that two separately nucleated lamellar crystals collided with each other. A thin part was present at the centre of the lamella, implying that

the upper and lower lamellae did not join perfectly, owing to the chain misorientation between them.

The locational relationships between the lamellae can then be considered. Figure 3g shows non-stacked lamellae (hereafter referred to as 'isolated lamellae'; 1, 2, and 3) and stacked lamellae (4). Isolated lamellae 1, 2, and 3 exhibited chain orientation angles of approximately 60°, 175°, and 40°, respectively, indicating that their chain orientations differed significantly. In contrast, the upper three parts in stacked lamella 4 exhibited similar chain orientation angles (with differences of <10°). These stacked lamellae with similar chain orientations indicated that the lamellae branched from the others[46] and/or were linked with tie chains[47].

Figure 4a shows a map of the chain tilt angle $\varphi$ (the angle between the normals to the flat faces and chains in the lamellae). Figure 4b shows a histogram of $\varphi$ measured for all the analysed lamellae (in Fig. 4 and Supplementary Figs. 12 and 13), which exhibits a broad peak with the top at ~15°. Nearly 90 % of the analysed pixels are contained

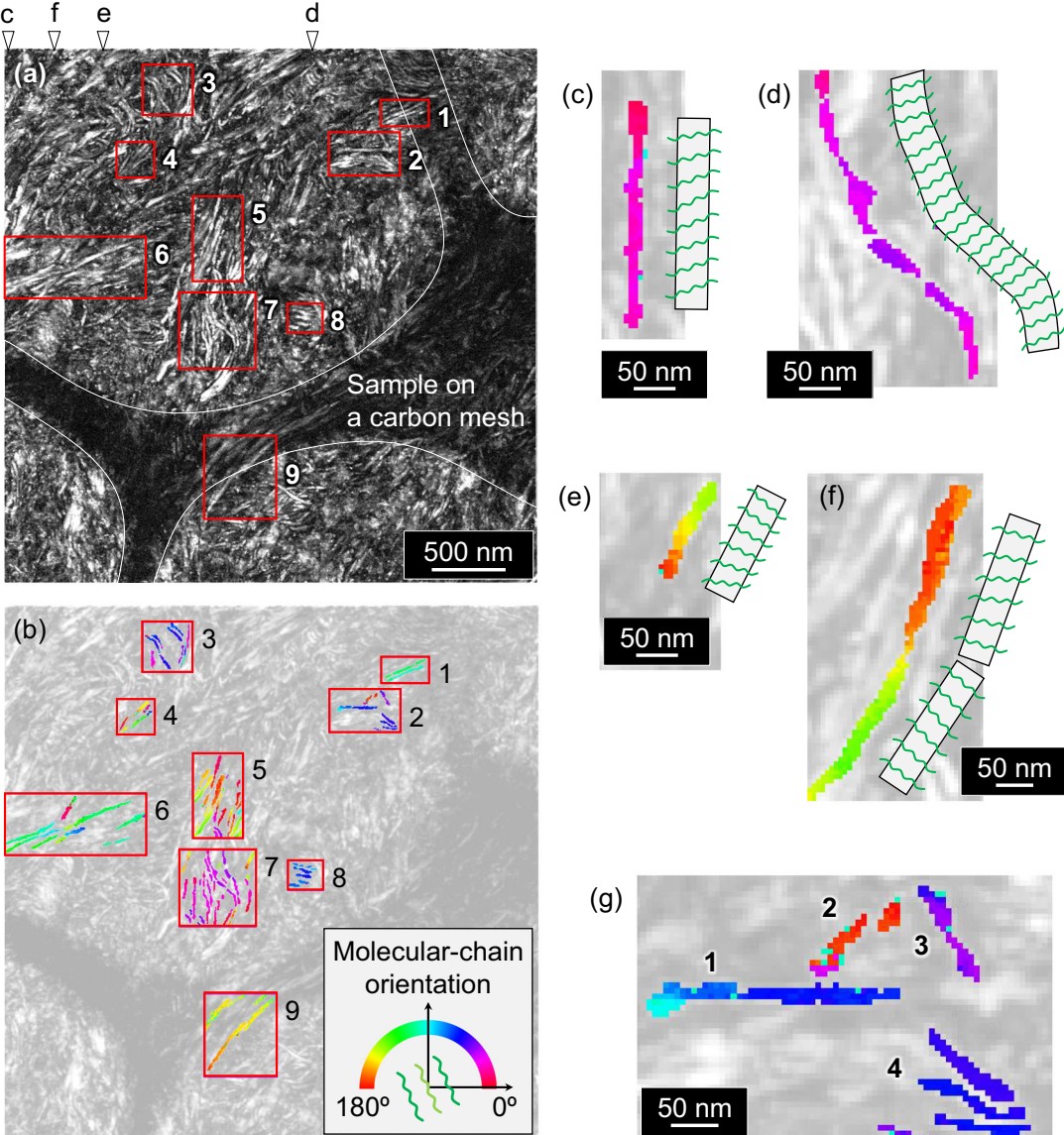

**Fig. 3 | Analysis of molecular chain orientations. a** Reconstructed dark-field (DF)-STEM image of HDPE. White triangles mark the locations from which the ED patterns shown in Fig. 2c–f were extracted, and red frames denote the analysed regions. **b** Map of the chain orientation angles in the Cartesian coordinate system (absolute molecular-chain orientations). **c** Straight and **d**, curved lamellae with a uniform chain orientation extracted from the left and central parts of Region 7 in (**b**), respectively. **e** Straight lamella with gradually changing chain orientations extracted from the upper-right part of Region 7 in (**b**). **f** Lamellae with two different chain orientations extracted from the central part of Region 5 in (**b**). **g** Lamellae corresponding to Region 2 in (**b**), of which 1, 2, and 3 are nonparallel, whereas 4 is stacked.

between $\varphi = 0°$ and 30°, and there are few pixels (3 %) above 50°. The $\varphi$ corresponding to the peak top is significantly smaller than the thermodynamically favoured $\varphi$ of 34°[13,14,22,23].

Next, the origins of the variation in $\varphi$ were considered. Figure 4c–e show maps of the chain tilt angle for isolated lamellae, with no parallel lamellae within 30 nm, extracted from Regions 2, 5, and 7 of Fig. 4a, respectively. The representative chain tilt angles in the lamellae in Fig. 4c, the upper part of the lamella in Fig. 4d, and the left and right lamellae in Fig. 4e were 31°, 25°, 28°, and 30°, respectively. The isolated lamellar crystals exhibited chain tilt angles close to the thermodynamically favoured value ($\varphi = 34°$). These results indicated that the annealing condition used in the present study was sufficient to align the chains in isolated lamellae. The lower part of the lamella in Fig. 4d was considered a stacked lamellar region and exhibited an average chain tilt angle of 8°. Figure 4f–h show maps of the chain tilt angle for the stacked lamellae extracted from Regions 1, 2, and 5 of Fig. 4a, respectively. In the case of Fig. 4f, the representative chain tilt angles in

the upper and lower lamellae were 5° and 15°, respectively. Figures 4g, h show similar tendencies (1°–23°). Therefore, the chain tilt angles for stacked lamellae were smaller than the thermodynamically stable value of 34° (Supplementary Figs. 12 and 13). These small tilt angles would also be preferred values for the stacked lamellae in the present specimen.

Figure 4i shows histograms of $\varphi$ for the isolated and stacked lamellae. We note that all the lamellae shown in Fig. 4 and Supplementary Figs. 12 and 13 are included in the histograms of Figs. 4b and i. The histogram of the isolated lamellae exhibits a high frequency of tilt angles close to the thermodynamically stable value of 34°, while that of the stacked lamellae has peak positions unambiguously smaller than 34°. Because the ratio of the number of stacked lamellae to the total number of lamellae analysed was ~94%, the integrated histogram of the isolated and stacked lamellae in Fig. 4b exhibited a single peak at -15°.

The reasons for the isolated and stacked lamellar crystals exhibiting different chain tilt angles are discussed. As described

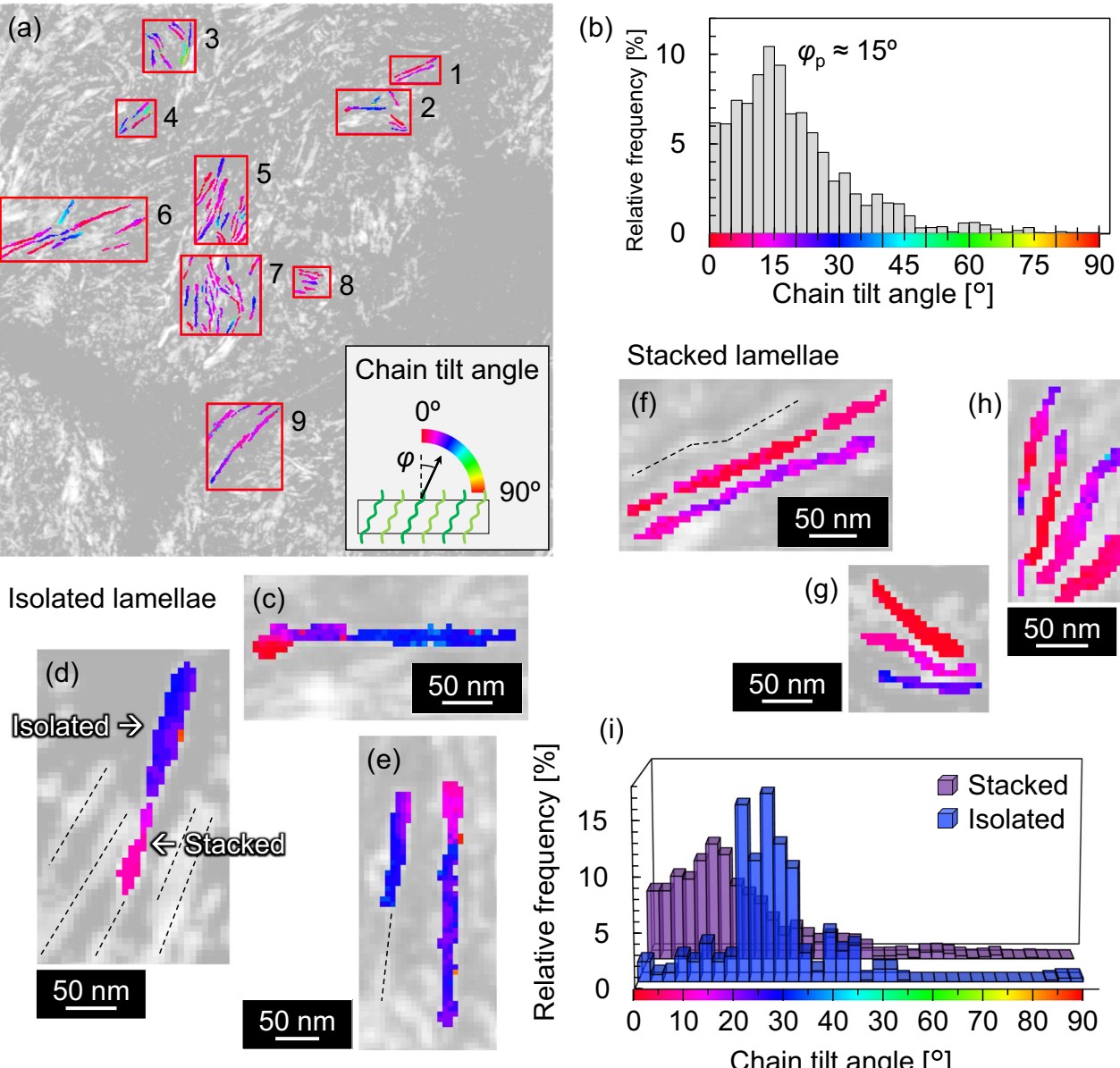

**Fig. 4 | Analysis of chain tilt angles. a** Map of the chain tilt angle $\varphi$ with respect to Fig. 3a, b. **b** Histogram of $\varphi$ for all the lamellae shown in (**a**) and Supplementary Figs. 12 and 13. The histogram was normalized by the total area of the lamellae analysed for the determination of $\varphi$. The broad peak exhibits a peak top at -15°. **c–e** Isolated lamellae from Regions 2, 5, and 7, respectively, with large $\varphi$ values (close to 34°). **f–h** Stacked lamellae from Regions 1, 2, and 5, respectively, with small $\varphi$ values (<34°). **i** Histograms of $\varphi$ for isolated and stacked lamellae. The histogram for isolated lamellae was normalized by the total area of the isolated lamellae, and that for stacked lamellae was normalized by the total area of the stacked lamellae. Because stacked lamellae were dominant in the present specimen, the histogram for all lamellae shown in (**b**) exhibits a feature similar to that for the stacked lamellae.

above, amorphous chains are arranged near the flat faces of the lamellae to moderate the density anomaly, considering the three effects of chain ends, sharp folds, and chain tilt[14]. Because our specimen had a high molecular weight (weight-averaged molecular weight $M_w \geq 10^5$ g/mol, estimated from the low melt flow rate of 0.06 g/10 min[48]), the fraction of the chain ends in the specimen was small. The moderation of the density anomaly due to the chain ends was negligible[14]. Thus, the fraction of sharp folds near the flat faces of the lamellae was presumably the only factor affecting the chain tilt in the lamellae. Theory[48] and simulations[13] predicted that the sharp-fold fraction decreases with an increase in the tilt angle. Fritzsching et al. used the sharp-fold fraction of ~1/3 and the thermodynamically stable $\varphi$ of 34° to moderate the density anomaly near lamellar surfaces[14]. Our experimental result, i.e., smaller $\varphi$ values in stacked lamellae than in isolated ones, implies that the sharp-fold fractions may differ between stacked and isolated lamellae.

Nanoscale analysis revealed that two types of lamellar crystals (isolated and stacked) were present in a single specimen and that they exhibited different chain tilt angles (Fig. 4i). The reason for such different tilt angles remains an open question and is expected to be revealed by simulating the crystallisation dynamics near the lamellar surface with the knowledge of the directly measured tilt angles of all the lamellae examined in the present study.

In conclusion, lamellar crystals in an HDPE specimen prepared via short-duration annealing were successfully visualised through NDI, a state-of-the-art electron-staining-free electron microscopy technique. Numerous position-resolved ED patterns were used to evaluate the molecular-chain orientations in individual lamellae. Precise analysis of the relationships between the morphologies and inner-chain

orientations revealed various types of lamellar crystals: (i) straight lamellae with uniform chain orientations, (ii) curved lamellae with uniform chain orientations, (iii) straight lamellae with gradually changing chain orientations, and (iv) lamellae with multiple parts having different chain orientations. While isolated lamellae exhibited $\varphi$ values close to the thermodynamically favoured angle (34°), stacked lamellae predominantly had $\varphi$ values smaller than 34°. It is worth noting that the observations made in this study (particularly the histogram of $\varphi$) depend on the primary structure and thermal history of the analysed specimen and can vary with respect to the crystallisation conditions. For example, well-annealed HDPE specimens have exhibited roof-like lamellae with only one chain tilt angle of ~34°[22,23]. Because stacked and isolated lamellar crystals would exhibit different physical properties owing to their differing chain tilt angles, the properties of PE can be readily tuned by controlling the amount and ratio of these lamellae in an appropriate preparation method. Low-dose NDI can be applied to semicrystalline polymers because most of them are less susceptible to electrons than HDPE[49]. Precise control of the structure and properties of semicrystalline polymers will curtail the weight and utilisation of polymeric materials.

## Methods

### Materials

HDPE pellets (Novatec HD HF310; Nippon Polychem; density = 0.950 g/cm³; melt flow rate = 0.06 g/10 min; weight-averaged molecular weight ($M_w$) ≥ 10⁵ g/mol[50]) were processed into a 0.5-mm-thick film via hot pressing. The pressed film was cut into a 20-mm square, and then melted on a hotplate at 160 °C for 5 min. The melted film was then quickly moved to another hotplate set to 120 °C and kept there for 1 h to allow crystallisation. After the thermal annealing, the film was rapidly cooled on a metal surface precooled with liquid nitrogen to freeze the structure. Approximately 1 mm of the outer edge of the film was cut off, and the central part was used as the sample in a series of measurements. Therefore, the lamellar crystals in the sample were assumed to be randomly oriented.

### WAXD and SAXS measurements

Simultaneous WAXD and SAXS measurements were performed at the beamline BL-6A ($\lambda$ = 0.1500 nm) of the Photon Factory at KEK (Tsukuba, Japan). A beam size of 200 μm × 500 μm in the vertical and horizontal directions, respectively, was used. 2D WAXD and SAXS patterns were acquired using PILATUS 100 K and PILATUS3 1 M detectors (DECTRIS Ltd.), respectively. Silver behenate was used as the standard. The 2D WAXD and SAXS patterns were azimuthally averaged to produce one-dimensional (1D) profiles of scattering intensity vs. $q$, where $q$ is the scattering vector ($q = 4\pi \sin(\theta)/\lambda$; $2\theta$: scattering angle [°]). A laboratory-scale WAXD apparatus (D8 ADVANCE, Bruker) equipped with a 2D detector (PILATUS3 R 100K-A, DECTRIS Ltd.) was used to evaluate the degree of crystallinity and orientation because it had a wider $q$ range than the WAXD system used in the simultaneous synchrotron measurements. A mirror-monochromatic X-ray beam ($\lambda$ = 0.1542 nm) was collimated to a diameter of 500 μm immediately before it reached the sample. Silicon powder was used to calibrate the sample-to-detector distance and $q$. The contribution of air scattering to the 1D profiles was eliminated by considering the X-ray transmittance of the specimen.

### Preparation of ultrathin sections for TEM and NDI

An EM UC7 Ultramicrotome (Leica Microsystems) was used to prepare ultrathin (~100 nm) sections along the film thickness direction under cryogenic conditions (−155 °C), which were then collected on a carbon-mesh microgrid with pore diameters of a few micrometres. All the ultrathin sections were coated with ~5-nm-thick carbon to reduce the electron irradiation damage. Prior to the TEM

observations, certain microtomed sections were stained with $RuO_4$ at 300 Pa for 2 h using a vacuum electron stainer (Filgen, Inc.). All the ultrathin sections were coated with ~5-nm-thick carbon to reduce the irradiation damage.

### TEM and STEM observations

TEM and STEM investigations were performed using a JEM-F200 instrument (JEOL) at an acceleration voltage of 200 kV with conventional STEM and pixelated detectors; the former were BF and ADF detectors (both manufactured by JEOL), and the latter were OneView and K3 IS cameras (both manufactured by Gatan).

ED patterns of unstained ultrathin sections were recorded using the K3 IS camera, with a beam diameter of 2.1 μm. A gold single crystal was used as a standard sample for camera-length calibration.

BF-TEM images were recorded using the OneView camera to analyse the $RuO_4$-stained ultrathin sections of crystallised HDPE. Because $RuO_4$ preferentially stains the amorphous region, the bright and dark regions in BF-TEM images correspond to crystalline- and amorphous-rich regions, respectively[32,33].

A 2D scan for NDI was performed using an electron beam with an $\alpha$ value of 1.3 mrad and a calculated diameter of 1.2 nm (full width at half maximum [FWHM])[35,36]. The ED pattern at each point was recorded using the K3 IS camera (256 × 256 pixels, half of the detector area with 8× binning). A gold single crystal was used as a standard sample for camera-length calibration. The scan step was set to 6 nm, which ensured that both the beam and step sizes were smaller than the $l_c$ (17.8 nm) and $l_a$ (12.5 nm) obtained via SAXS. The exposure time was set to 0.0067 s per step (150 fps). Numerous ED patterns ($600^2 = 360{,}000$) were acquired and further analysed using a home-made Python script[37] to construct virtual DF-STEM images and determine the nanoscale chain orientation. Each ED pattern was subjected to several image-treatment schemes, including background subtraction, noise reduction, and peak extraction, to sharpen the diffraction peaks (Supplementary Fig. 5).

For electron-irradiation-sensitive materials, including semicrystalline polymers, the number of incident electrons per unit time and area (dose rate [e⁻/(Å² s)]) and the total number of incident electrons per area (dose [e⁻/Å²]) are essential parameters that ensure the validity of measurements. The intensity [e⁻/s] of the electron beam in the vacuum region was measured using the K3 IS DED camera, and the dose per various unit areas was calculated, as described below. In general, the electron beam in STEM has a Gaussian intensity distribution. Thus, the FWHM (=2.355$\sigma$, where $\sigma$ represents the standard deviation of the Gaussian distribution) was used as the beam diameter, which was evaluated as 1.2 nm from $\alpha$[35]; consequently, $\sigma$ was calculated to be 0.52 (=1.2/2.355) nm. When the FWHM was used to calculate the unit area, the calculated dose was 800 e⁻/Å². However, the area outside the FWHM in the Gaussian distribution was ~30%, which contributed significantly to electron irradiation damage. When the width, including 99.99% of the area (8$\sigma$ = 4.1 nm), was used to calculate the unit area, the dose was 70 e⁻/Å². Furthermore, the dose per scan step area ($6^2$ nm) was 26 e⁻/Å².

A larger $\alpha$ yielded a smaller beam size but a higher electron dose. To resolve this dilemma, the sample temperature was reduced to −175 °C using a cryo-holder (Gatan) with liquid nitrogen for reducing electron irradiation damage. A 50% reduction in the electron irradiation damage was expected with cooling from 23 to −175 °C, according to our damage evaluation of the same HDPE sample (unpublished work). The conditions for the NDI measurements were determined by balancing the electron-beam damage, ED pattern quality, and reconstructed-image resolution.

Conventional BF- and ADF-STEM images were obtained under the same optical conditions that were adopted for NDI. The detection-angle ranges for the BF and ADF detectors were 0–6.8 and 16.9–61.8 mrad, respectively.

## Data availability
The source data that support the findings of this study are available from the corresponding author upon request.

## Code availability
The code used for the analysis of the data in this study is available from the corresponding author upon request.

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

## Acknowledgements

This study was financially supported by the JST CREST program to H.J. (JPMJCR1993 and JPMJCR19T4), a JSPS KAKENHI grant to H.M. (JP20H02782), and IMRAM grants to H.M. (2019, 2020, 2021, and 2022). The authors thank Dr. H. Matsumoto and Dr. K. Mori (Mitsubishi Chemical Corporation) for providing PE pellets and thank Dr. E. Okunishi and Mr. K. Yamazaki (JEOL) for their advice on the NDI measurement conditions. The synchrotron WAXD and SAXS measurements were performed with the approval of the Photon Factory Advisory Committee (nos. 2019G112 and 2021G118).

## Author contributions

H.J. and S.K. conceived the research and designed the experiments. S.K. performed the experiments and analysed the data. S.K., H.M., and T.M. co-wrote the manuscript. All authors contributed to the discussion of the results and the text.

## Competing interests

The authors declare no competing interests.
