## [Peer Review File · Nature Communications]

Reassessing chain tilt in the lamellar crystals of polyethyleneReviewers' Comments:

Reviewer #1:

Remarks to the Author:

Nature Communications manuscript NCOMMS-23-11692: "Reassessing chain tilt in the lamellar crystals of semicrystalline polymers" by S. Kanomi et al. (Corresponding author: H. Jinnai)

The present contribution illustrates quite spectacularly the impact of technical improvements (here in electron microscopy, electron diffraction and, of course, image acquisition and processing) on our analysis of complex morphologies. Combining positional resolution of the electron beam with high-performance cameras makes it possible to analyse at a nanometer resolution the crystal structure and chain tilt at each location of individual chain folded lamellae, even when embedded in a less organized spherulite.

The work presented illustrates this progress in a very convincing way. The dark field images provided are stunning. The writing and the figures are very pedagogic, and do not hesitate to recall old issues and controversies in polymer science to justify the investigation. As such, the paper is at the level of previous contributions of this leading team in the field.

There are a few weaker points that must be mentioned. (a) The aim of the work – determining the tilt of chains in crystalline lamellae of polymers, and more precisely in the much-investigated polyethylene, is relatively limited (or highly focused). Much is known on this topic: the preference for relatively "simple" (in terms of Miller indices) fold surfaces that, in bulk polymers are usually $\{10l\}$ planes with $l=0, 1, 2$ and even 3 (chain tilts ranging from 0° up to 45° , all of which are duly supported by experimental evidence). One of the novel elements brought by this investigation is the dual existence of two of these tilts in different lamellae of one sample, and, in some cases, are shown to differ in the two sides/halves of the growing lamella. (b) The approach is not totally reliable. Some lamellae or lamellar parts are seen as diffracting when imaged in both 200 and 110 reflections (extended data Figure 6). The "overlap" of reflections mentioned does not allow defining which reflection is imaged. The structural information is in these cases ambiguous, since a (101) fold surface would yield diffraction spots at 18 or 32° (extended figure 8). In the very well developed and pedagogical supplementary information, the authors indicate however that in such cases, the data are discarded (c) Details of the sample preparation would be useful, as they may interfere with the results. Thin film growth has been privileged in earlier analyses dealing with these topics: they allow some simplifying assumptions. Relatively thin films were also used here, but "sections" are investigated. What sections, and how are they oriented relative to, for example, the growth direction? The authors indicate that parallelism of the chain axis with the e- beam may be assumed. The variety of lamellae orientations in such small areas does not correspond to the standard view, conveyed by e.g. polarized optical microscopy. This variety could be another novel element of the spherulite morphology. Conversely, similar stem orientations in neighbour, parallel lamellae are associated with traversing molecules, when a more standard cause would be screw dislocations (d) The study deals with one sample crystallized at one temperature. The material and the crystallizations conditions are different from the earlier studies, which limits the strength of the comparisons and the attempted generalizations (although, admittedly, a very basic feature is investigated) (e) As a final note, a remark and an advice: the claims mentioned about the possible impact of this study, are, curiously, not the usual mark of the major author. Although such claims have become quasi-necessary by today's standards (or imposed by funding agencies?), the chain tilt in polyethylene lamellae may, most probably, not much "influence the marine microplastic formation, alleviate polymer-related environmental issues", "diminish fossil fuel utilization" or "help realize a sustainable society". The mark of scientific contributions published in Nature resides to a significant extent in their controlled and sober language (this should be, but is not, the rule everywhere else). In this context the reference remains Watson and Crick's "It has not escaped our notice..." when announcing the DNA double helix structure, in Nature, 70 years ago. Other times...

To conclude: the work presented here is a technical prowess made possible by improvements of the electron microscopes and cameras. The overall feeling is that this work was designed to illustrate this progress and its applicability to unstained polymers. The dark field images are indeed spectacular and

novel. The scientific issues tackled appear to stem from the capabilities offered by the technique. The sample used has no special characteristic, and was probably designed to be that way. The connections (via identical or different tilt angles of the stems) between neighbour lamellae are established on a variety of samples, with a critical and very honest evaluation of their validity – a tribute to the technical skills and the ethics of the authors. It remains that much simpler experiments on specifically designed samples (thickness, thermal treatments, etc.) have provided most of the basic concepts in the field. The present work's value is to extend these insights to a "real world" case and sample. The price to pay for this increment of knowledge (investment, research effort, computing with the analysis of 600x600 images) may appear rightly as very (too?) high.

Bernard Lotz

Reviewer #2:

Remarks to the Author:

A modern electron-microscopy and nanodiffraction technique without staining was utilized here to determine chain tilt in polyethylene crystallites with very high spatial resolution. Figure 3 is quite impressive and convincing in terms of the lamellae observed. I found this paper interesting, even though I'm not fully convinced of the conclusions, found several details in the experiment and analysis unclear, and feel that it is forced and artificial to claim (line 51) that this fundamental work will help to "alleviate polymer-related environmental issues".

The description of the analysis of chain tilt from the microdiffraction data is not detailed enough. For instance, it leaves me wondering if a tilt contribution by chains out of the plane of the image has been overlooked. In Figure 4, particularly parts (b) and (d), the authors seem to make the assumption that the chains throughout both lamellae lie exactly in the plane of the paper. This would seem to be a favorable coincidence that needs to be remarked upon. In the supporting information, the authors do write that "ultrathin specimens for transmission electron microscopy (TEM) observation were carefully prepared so that the chain axis (c-axis) is normal to the electron beam." This procedure needs to be explained in the main text, and the spread of angles from the normal (90 degree) orientation needs to be estimated. Is it really true that all chain axes in Figure 3 and Fig. 9, in all the different lamellae and with the indicated wide range of different chain tilt angles, were exactly perpendicular to the electron beam?? If the chains were tilted relative to the plane of the paper, the visible tilt would be smaller than the actual tilt, which would reduce the discrepancy with the literature. One needs to consider what happens to chains that were not parallel to the sample surface, when they reach that surface. Are they cut or forced to lay flat?

As the authors know (line 67), at the end of lamellae the driving force for chain tilt is reduced. It seems that their analysis implicitly selects the ends of lamellae at the surface of the film. Just like the reconstructed surface of a crystal is different from the bulk, the lamellae near the surface here may have a different tilt than those in the bulk.

The initial framing around the claim (line 20) that the tilt angle values "have been disputed for decades" does not seem quite appropriate. There is actually a general consensus in the literature, which is later acknowledged (in line 232, referring to the "commonly accepted value of 35 degrees"). It is the smaller values reported here that may stir up a dispute.

Clarifications needed:

In the figure captions, "of PE" needs to be replaced with a more detailed brief description of the PE. The approximate number-average molecular weight needs to be specified – as the authors have implied, e.g. in line 67, at lower MW, with more chain ends, the small tilt angles found here would be less surprising. Similarly in the text, in line 156, these characteristics of "a PE film" need to be specified. What is the degree of branching?

In the caption of Figures 2 c-e, it is stated that "the direction of the incident electron beam is indicated by [uvw]" but numbers in square brackets are shown only in (e). What is [uvw] for (c) and (d)? And could the locations of c, d, and e please be marked in (b), and in Figure 3?

Line 180: "chain orientation being vertical to the two-peak connecting line." What does "vertical to" mean? Please rephrase for more clarity.

Line 213: "chain tilt from the perpendicular c-axis orientation" also is unclear: the chain is along the c-axis, so how can there be "tilt from" it? This needs to be rephrased for clarity.

The title "Reassessing chain tilt in the lamellar crystals of semicrystalline polymers" seems too general for a study of one sample of polyethylene. "Reassessing chain tilt in the lamellar crystals of semicrystalline polyethylene" would be more accurate.

Reviewer #3:

Remarks to the Author:

This communication reports the application of a fairly sophisticated scanning transmission electron microscopy (STEM) technique dubbed "nanodiffraction imaging" (NDI) to study the tilt of chain stems within lamellar crystals of polyethylene (PE) crystallized from the melt. The new thing is apparently the very fine beam of electrons that allows a lateral resolution as small as 1-2 nm, much finer than conventional beams of X-rays, electrons or neutrons, which are on the order of micrometers. Using this high resolution beam, the authors report the analysis of electron diffraction patterns from single lamellae of polyethylene in ultrathin sections (100 nm thick) obtained by cryomicrotomy of melt-crystallized thin films.

The technique is valuable and its application to the examination of tilt angles in PE is interesting. Other aspects of the presentation, however, are bothersome from the polymer science perspective, and tend to overstate the significance of the result. A few of these aspects are highlighted below.

(1) The authors seem to imply incorrectly that tilt angle in semicrystalline polymers is an inherent feature, without regard to chemistry or sample history. Many semicrystalline polymers do not exhibit a clear-cut lamellar morphology at all. Both the title and the opening paragraph should be revised to mention PE, the polymer studied in this work, explicitly.

(2) Even in those polymers, like PE, that do exhibit lamellar crystal morphologies, the lamellar morphology is highly dependent on the source of PE (how synthesized, molecular weight, etc.) and how the sample was crystallized (whether from solution or melt, at what temperature, cooling rate or thermal history, and whether flow was involved). Thus, it is not necessarily surprising that samples of semicrystalline polyethylene prepared in different labs under different conditions exhibit different tilt angles.

(3) Recognizing that tilt angle is sensitive to crystallization conditions, it is somewhat of an overstatement to represent discrepancies in reported values as some great dilemma. Unless all aspects of sample chemistry and history are controlled, the reporting of different tilt angles by different labs (such as refs 8,9 and ref 10) is revealing, but neither is necessarily "wrong". Even in this work, where the peak values are close to 5 or 15 deg, the range of tilt angles reported is quite large (up to 70 deg). Thus, assertions like (p15) "the predominant phi value in the specimen was experimentally proven to be 15° herein, which differs from the commonly accepted value of 35°" come across as misleading and argumentative. After all, how can the value of 35° be both "commonly accepted" and "disputed for decades" (p2)?

(4) There is one value of tilt angle that might be special: the thermodynamically most favored one.

The discussion of the increase of tilt angle with increasing T_c (p15,16), reported by ref 10 and also in this work, is consistent with a larger tilt angle being thermodynamically favored, as also argued in ref 13. It suggests that the "proven" value of 15 deg reported here is probably a consequence of the prevailing crystallization kinetics during sample preparation. At the very least, the authors should provide sufficient detail about the PE used, film preparation and thermal history in the main text or SI that the crystallization conditions could be reproduced. The single sentence provided on p10 is not enough.

(5) On p9, the authors assert that their "precise chain-tilt analysis resolves the 40-year-old dilemma of linking the structural nature of polymer crystals to their physical properties." If referring to optical, thermal or mechanical properties mentioned in p3, what is the new "linkage" to those properties that NDI provides? There is no real debate in the polymer community that properties depend on structure, but how those properties depend specifically on tilt angle could be better articulated. The assertion that the current work will somehow have measurable impact on marine microplastics seems particularly tenuous.

In addition, clarification of the following points is recommended.

(6) Assuming that the diffraction patterns are collected in transmission mode (Fig 2), the resolution in the thickness direction would seem to be controlled by the sample thickness (~ 100 nm), which is much larger than the lamellar thickness but smaller, perhaps, than the lamellar width. If the latter is not the case, then I would expect the ED pattern to comprise multiple lamellae (like Fig 2f). Some mention in the main text about the sample thickness used and the restriction imposed by the condition on lamellar width is warranted.

(7) P12: The phrase "chain orientation being vertical to the two-peak-connecting line" is unclear. Do the authors mean "perpendicular"? Also, "line connecting the two peaks" would be better than "two-peak-connecting line".

(8) All of the Extended Data figures appear to be referenced by the wrong figure numbers in the SI. It might also be the case for Extended Data figures referenced in the main text. This mistake is a significant hindrance to reviewing the paper. Authors should check the figure numbering carefully.

(9) Extended Data Fig 8 caption refers to "parallel and antiparallel chains of the orthorhombic PE." However, unlike polypropylene, the PE chains do not have a unique direction; the parallel/antiparallel distinction is meaningless. (It is the setting angle that distinguishes the two chains in the unit cell.)

Replies to Reviewer #1 (Dr Bernard Lotz)

The present contribution illustrates quite spectacularly the impact of technical improvements (here in electron microscopy, electron diffraction and, of course, image acquisition and processing) on our analysis of complex morphologies. Combining positional resolution of the electron beam with high-performance cameras makes it possible to analyse at a nanometer resolution the crystal structure and chain tilt at each location of individual chain folded lamellae, even when embedded in a less organized spherulite.

The work presented illustrates this progress in a very convincing way. The dark field images provided are stunning. The writing and the figures are very pedagogic, and do not hesitate to recall old issues and controversies in polymer science to justify the investigation. As such, the paper is at the level of previous contributions of this leading team in the field.

→ Dr Lots, we greatly appreciate your careful reading of our manuscript and your valuable comments, which were very helpful in strengthening our manuscript. We have responded to your queries/comments below. Please note that the modifications in the revised manuscript have been highlighted in red.

(a) The aim of the work – determining the tilt of chains in crystalline lamellae of polymers, and more precisely in the much-investigated polyethylene, is relatively limited (or highly focused). Much is known on this topic: the preference for relatively “simple” (in terms of Miller indices) fold surfaces that, in bulk polymers are usually $\{10\}$ planes with $l=0, 1, 2$ and even 3 (chain tilts ranging from 0° up to 45° , all of which are duly supported by experimental evidence). One of the novel elements brought by this investigation is the dual existence of two of these tilts in different lamellae of one sample, and, in some cases, are shown to differ in the two sides/halves of the growing lamella.

→ We thank the reviewer for this thoughtful comment, as it has made us reconsider our study's novelty, that is, directly identifying the crystal structure and chain tilt at each location of the individual chain-folded lamellae in an HDPE specimen. This identification strategy permits detailed visualisation of the lamellar structures and elucidation of the formation mechanisms. Consequently, we have amended the focus of our study from determining the major chain tilt in the specimen to (i) visualising the nanoscale distribution of both lamellar crystals and chain orientations, and (ii) clarifying the formation mechanisms of the lamellar crystals at each location.

To reflect these points, we have made significant changes to our manuscript, as follows:

Page 2, Lines 21–28

Before: Here we report the direct determination of molecular chain orientations in lamellar crystals using a novel electron-diffraction-based imaging technique with nanometre-scale positional resolution¹⁴ and provide compelling evidence for the existence of diverse ϕ values (primarily $\sim 15^\circ$). Greater clarification of the nanoscale structure–property relationships of semicrystalline polymers can permit precise tuning of their properties and pave the way for lightweight resource-saving material design.

After: Here, we report the direct determination of molecular chain orientations in the lamellar crystals of high-density PE using a novel electron-diffraction-based imaging technique with nanometre-scale positional resolution and provide compelling evidence for the existence of lamellar crystals with different inner-chain orientations. Factors dictating the ϕ values of lamellae were determined. Clarifying the nanoscale variation in lamellar crystals and their formation mechanisms in PE can permit precise tuning of properties and expedite resource-saving material design.

Page 8, Lines 121–128

Before: In this study, the nanoscale chain orientation in lamellar crystals was “directly” observed by NDI. Chain tilt was experimentally evaluated without assumptions by analysing numerous position-resolved ED patterns. This precise chain-tilt analysis resolves the 40-year-old dilemma of linking the structural nature of polymer crystals to their physical properties.

After: In this study, the nanoscale morphologies of lamellar crystals and the molecular chain orientations in lamellae were visualised simultaneously in a ‘direct’ manner using position-resolved ED patterns. Chain tilt in lamellae was experimentally identified without any assumptions through orientational relationships between the lamellae and their inner-chain orientations. Precise analysis revealed the variation in the lamellar morphology and the chain tilt inside lamellae for an HDPE specimen prepared by short-duration annealing. Furthermore, mechanisms governing chain tilt in lamellae are proposed.

In addition to these corrections, we have significantly revised the Discussion and Conclusion sections of the manuscript. However, in the interest of brevity, these extensive changes have not been listed in this response letter; please see pages 11–17 of the manuscript, in which the changes are shown in red. Additionally, we have modified Figs. 3 and 4, as follows:

Fig. 3 | **a**, Reconstructed dark-field (DF)-STEM image of HDPE. White triangles mark the locations from which the ED patterns shown in Figs. 2c–f were extracted, whereas red frames denote the analysed regions. **b**, Map of chain orientation angles in the Cartesian coordinate system (absolute molecular chain orientations). **c**, Straight and **d**, curved lamellae with a uniform chain orientation extracted from the left and central parts of Region 7 in **(b)**, respectively. **e**, Straight lamella with gradually changing chain orientations extracted from the upper-right part of Region 7 in **(b)**. **f**, Lamellae with two different chain orientations extracted from the central part of Region 5 in **(b)**. **g**, Lamellae corresponding to Region 2 in **(b)**, of which 1, 2, and 3 are non-parallel, whereas 4 is stacked.

Fig. 4 | a, Map of chain-tilting angle φ with respect to Figs. 3a and b. **b**, Histogram of φ for all analysed lamellae. The broad peak exhibits a peak top at $\sim 15^\circ$. **c,d**, Stacked lamellae from Regions 1 and 5, respectively, with small φ values ($< 15^\circ$). **e,f**, Isolated lamellae from Regions 2 and 7, respectively, with large φ values (close to 35°).

(b) The approach is not totally reliable. Some lamellae or lamellar parts are seen as diffracting when imaged in both 200 and 110 reflections (extended data Figure 6). The “overlap” of reflections mentioned does not allow defining which reflection is imaged. The structural information is in these cases ambiguous, since a (101) fold surface would yield diffraction spots at 18 or 32° (extended figure 8). In the very well developed and pedagogical supplementary information, the authors indicate however that in such cases, the data are discarded.

→ We thank the reviewer for this comment. In the original manuscript, we included a DF-STEM image reconstructed from the integrated intensities of the 110 and 200 spots in Fig. 3 because it showed the internal structures with better contrast. However, as indicated by the reviewer, the image with mixed intensities was not appropriate for our discussions. As explained in the Supplementary Information (SI), chains in the lamellae were oriented in the in-plane

direction (perpendicular to the plane of the image) when the lamellae were visualised with the 200 reflection and viewed from the edge-on direction. Therefore, we have moved the DF-STEM image reconstructed using *only 200 spot* intensities (Extended Data Fig. 6b in the previous manuscript) to Fig. 3a in the revised manuscript. (In accordance with the journal guidelines, we have moved all Extended Data figures to the SI file and renamed and cited them as Supplementary Figures). It is worth noting that in the original manuscript, the chain orientation angles and chain-tilting angles were estimated using only the ED patterns with 200 spot intensities. In this regard, we have revised the manuscript as follows:

Page 7, Lines 107–111 (Supplementary Information)

Before: Thus, the above process was able to separate the 110 and 200 intensities.

After: Thus, the aforementioned process separated the 110 and 200 reflection intensities through a hexagonal-like pattern.

For each scanning position (x, y), azimuthal profiles were created in the same manner as described in the previous section. The maximum intensities of the 200 reflection profiles were calculated and plotted to obtain the reconstructed dark-field scanning TEM (DF-STEM) image shown in Fig. 3a.

Page 10, Lines 164–166

Before: Subsequently, dark-field (DF) STEM images were reconstructed using the two highest-intensity centre-symmetric $hk0$ peaks in the image-processed ED patterns (Extended Data Fig. 5; details in Supplementary Information).

After: Subsequently, dark-field (DF) STEM images were reconstructed using the two highest-intensity centre-symmetric 200-reflection peaks in the image-processed ED patterns (Supplementary Fig. 5 and Supplementary Discussion).

(c) Details of the sample preparation would be useful, as they may interfere with the results. Thin film growth has been privileged in earlier analyses dealing with these topics: they allow some simplifying assumptions. Relatively thin films were also used here, but “sections” are investigated. What sections, and how are they oriented relative to, for example, the growth direction? The authors indicate that parallelism of the chain axis with the e- beam may be assumed. The variety of lamellae orientations in such small areas does not correspond to the standard view, conveyed by e.g. polarized optical microscopy. This variety could be another novel element of the spherulite morphology. Conversely, similar stem orientations in neighbour, parallel lamellae are associated with traversing molecules, when a more standard cause would be screw dislocations.

→ We appreciate the reviewer for providing such insightful feedback. In terms of sample preparation, we employed conventional hot-pressing and isothermal annealing for film fabrication and crystallisation, respectively. The resulting specimen was confirmed to be in a non-oriented crystallised state using X-ray diffraction/scattering techniques. Therefore, we validated the presence of lamellar crystals and chains with diverse orientations inside the specimen. Ultrathin sections were cut off from the inner part of the bulk specimen using an ultramicrotome (Supplementary Fig. 10). We searched and selected an appropriate field of view for evaluating the chain tilt (that is, most of the lamellae were nearly edge-on against the incident electron beam). The details concerning sample preparation have been added to the Methods section.

No spherulites were observed in our PE specimen through polarised optical microscopy after crystallisation. Although the results obtained in this study could not be related to spherulite structures, we are currently trying to perform a similar analysis on a specimen with clear spherulites.

Page 17, Lines 288–297

Before: PE pellets (Novatec HD HF310; Nippon Polychem; density, 0.950 g/cm³; melt flow rate, 0.06 g/10 min) were processed into 0.5-mm-thick films by hot pressing. The pressed films were melted at 160 °C for 5 min and further annealed at 120 °C for 1 h for crystallization. The lamellar crystals in the film were assumed to be randomly oriented.

After: HDPE pellets (Novatec HD HF310; Nippon Polychem; density = 0.950 g/cm³; melt flow rate = 0.06 g/10 min; weight averaged molecular weight (M_w) $\geq 10^5$ g/mol⁴⁸) were processed into a 0.5-mm-thick film by hot pressing. The pressed film was cut into a 20 mm square and then melted on a hot plate at 160 °C for 5 min. The melted film was then quickly moved to another hot plate set to 120 °C and maintained for 1 h to permit crystallisation. After the thermal annealing, the film was rapidly cooled on a metal surface precooled with liquid nitrogen to freeze the structure. Approximately 1 mm of the outer edge of the film was cut off, and the central part was used as the sample in a series of measurements. Therefore, the lamellar crystals in the sample were assumed to be randomly oriented.

Page 19, Lines 317–319

Before: An EM UC7 microtome (Leica Microsystems) was used to prepare ultrathin (~100 nm) sections of the crystallized films, which were produced under cryo-conditions (–155 °C) and collected on carbon-mesh microgrids with pore diameters of a few micrometres.

After: An EM UC7 microtome (Leica Microsystems) was used to prepare ultrathin (~100 nm)

sections **along the film thickness direction** under cryo-conditions ($-155\text{ }^{\circ}\text{C}$), which were then placed on carbon-mesh microgrids with pore diameters of a few micrometres.

We prepared a schematic portraying the preparation of the ultrathin sections (Supplementary Fig. 10).

Supplementary Fig.10 | Schematic describing the preparation of ultrathin sections. The central part (b) of a crystallised film (a) was cut off, and ultrathin sections parallel to the sides of the specimen were prepared (c).

(d) The study deals with one sample crystallized at one temperature. The material and the crystallizations conditions are different from the earlier studies, which limits the strength of the comparisons and the attempted generalizations (although, admittedly, a very basic feature is investigated)

→ We thank the reviewer for this essential comment. We agree with the point raised in that the major chain-tilting angle observed in our specimen cannot be generalised, because our HDPE specimen and crystallisation conditions are not necessarily comparable with those employed in previous studies. Therefore, as mentioned in our response to Comment (a), we have changed the focus of the paper from generalising the observed tilting angle to analysing and discussing the examined features and possible mechanisms for our HDPE specimen.

(e) As a final note, a remark and an advice: the claims mentioned about the possible impact of this study, are, curiously, not the usual mark of the major author. Although such claims have become quasi-necessary by today's standards (or imposed by funding agencies?), the chain tilt in polyethylene lamellae may, most probably, not much "influence the marine microplastic formation, alleviate polymer-related environmental issues", "diminish fossil fuel utilization" or "help

realize a sustainable society". The mark of scientific contributions published in Nature resides to a significant extent in their controlled and sober language (this should be, but is not, the rule everywhere else). In this context the reference remains Watson and Crick's "It has not escaped our notice..." when announcing the DNA double helix structure, in Nature, 70 years ago. Other times...

→ We appreciate the reviewer's constructive advice. We are very aware of the importance of revising the statement mentioned above. We included that assertion in the original manuscript because we thought that the thermodynamic stability of lamellar crystals, which is heavily influenced by the molecular chain tilt, is related to the decomposition of marine microplastics. Amorphous chains are believed to preferentially degrade during seawater- and sunlight-induced decomposition owing to an increase in crystallinity. The subsequent step of the degradation is the decomposition of the crystalline region. Generally, crystals with greater thermodynamic stability are less influenced by external stimuli such as heat and other forces. Hence, we presumed that more thermodynamically stable crystals would also be less affected by seawater and sunlight. Essentially, our opinion was that the decomposition of lamellar crystals could be controlled by tuning the chain tilt, thereby lowering the environmental impact of polymeric materials.

However, this significant leap in logic could not be explained in the manuscript. Furthermore, thanks to the reviewers' comments, we recognised that the thermophysical and mechanical properties deserved greater attention than the impact on marine microplastics. Therefore, we have removed the description of marine microplastics and included the following details:

Page 3, Lines 41–45

Before: Thus, the crystallization-dependent hierarchical structures of semicrystalline polymers (Extended Data Fig. 1) significantly affect their physical properties.

The fraction, size, and orientation of hierarchical structural elements, including lamellar crystals, higher-order structures²¹, chain conformation, and unit-cell packing²², significantly influence optical, thermal, and mechanical properties. For example, the enzymatic hydrolysis of crystallized chains is considerably slower than that of amorphous chains in spherulite-containing polyesters²³. These insights help elucidate the formation mechanism of marine microplastics²⁴. PE is the most mass-produced and mass-discarded plastic, and is predominantly responsible for marine microplastic pollution²⁵. Therefore, insights into the internal nanostructures of PE and their influence on the marine microplastic formation mechanism can alleviate polymer-related environmental issues.

After: Thus, the crystallisation-dependent hierarchical structures (Supplementary Fig. 1)

significantly affect the physical properties of the polymers. The fraction, size, and orientation of the hierarchical structural elements – including lamellar crystals, higher-order structures⁹, chain conformations, and unit-cell packing¹⁰ – significantly influence the optical, thermal, and mechanical properties.

Page 17, Lines 282–284

Before: Precise control of the structure and properties of semicrystalline polymers will reduce the weight and use of polymeric materials, diminishing fossil fuel utilization as well as marine microplastic production, thereby helping realize a sustainable society.

After: Precise control of the structure and properties of semicrystalline polymers will **curtail the weight and utilisation** of polymeric materials.

To conclude: the work presented here is a technical prowess made possible by improvements of the electron microscopes and cameras. The overall feeling is that this work was designed to illustrate this progress and its applicability to unstained polymers. The dark field images are indeed spectacular and novel. The scientific issues tackled appear to stem from the capabilities offered by the technique. The sample used has no special characteristic, and was probably designed to be that way. The connections (via identical or different tilt angles of the stems) between neighbour lamellae are established on a variety of samples, with a critical and very honest evaluation of their validity – a tribute to the technical skills and the ethics of the authors. It remains that much simpler experiments on specifically designed samples (thickness, thermal treatments, etc.) have provided most of the basic concepts in the field. The present work's value is to extend these insights to a "real world" case and sample. The price to pay for this increment of knowledge (investment, research effort, computing with the analysis of 600x600 images) may appear rightly as very (too?) high.

→ As indicated by the reviewer, we intended to demonstrate the power of the NDI technique for nanoscale analysis of semicrystalline polymers in this study. In particular, we have bolstered the revised manuscript not only with technical details, but also with substantial discussions based on the reviewers' valuable inputs. We believe that these revisions have significantly improved the quality of our manuscript.

Replies to Reviewer #2

A modern electron-microscopy and nanodiffraction technique without staining was utilized here to determine chain tilt in polyethylene crystallites with very high spatial resolution. Figure 3 is quite impressive and convincing in terms of the lamellae observed. I found this paper interesting, even though I'm not fully convinced of the conclusions, found several details in the experiment and analysis unclear, and feel that it is forced and artificial to claim (line 51) that this fundamental work will help to "alleviate polymer-related environmental issues".

→ We appreciate the reviewer's careful reading of our manuscript and their valuable comments, which have helped remarkably in strengthening our manuscript. In the following sections, we have answered the reviewer's questions/comments. It is worth noting that the modifications in the revised manuscript have been highlighted in red.

The details concerning the potential environmental benefits of our study have also been flagged by other reviewers. Therefore, we agree that the logic underlying these claims should be reinforced. We included these details in the original manuscript because we thought that the thermodynamic stability of lamellar crystals, which is heavily influenced by the molecular chain tilt, is related to the decomposition of marine microplastics. Amorphous chains are believed to preferentially decompose during seawater- and sunlight-induced degradation owing to an increase in crystallinity. The subsequent step of the degradation is the decomposition of the crystalline region. Generally, crystals with greater thermodynamic stability are less influenced by external stimuli such as heat and other forces. Hence, we speculated that more thermodynamically stable crystals would also be less affected by seawater and sunlight. Essentially, our opinion was that the decomposition of lamellar crystals could be controlled by tuning the chain tilt, thereby lowering the environmental impact of polymeric materials.

However, this significant leap in logic could not be explained adequately in the manuscript. Moreover, thanks to the reviewers' comments, we recognised that the thermophysical and mechanical properties deserved greater focus than the impact on marine microplastics. Therefore, we have deleted the description of marine microplastics and included the following details:

Page 3, Lines 41–45

Before: Thus, the crystallization-dependent hierarchical structures of semicrystalline polymers (Extended Data Fig. 1) significantly affect their physical properties.

The fraction, size, and orientation of hierarchical structural elements, including lamellar crystals, higher-order structures²¹, chain conformation, and unit-cell packing²², significantly influence optical, thermal, and mechanical properties. For example, the enzymatic hydrolysis of crystallized chains is considerably slower than that of

amorphous chains in spherulite-containing polyesters²³. These insights help elucidate the formation mechanism of marine microplastics²⁴. PE is the most mass-produced and mass-discarded plastic, and is predominantly responsible for marine microplastic pollution²⁵. Therefore, insights into the internal nanostructures of PE and their influence on the marine microplastic formation mechanism can alleviate polymer-related environmental issues.

After: Thus, the crystallisation-dependent hierarchical structures (Supplementary Fig. 1) significantly affect the physical properties of the polymers. The fraction, size, and orientation of the hierarchical structural elements – including lamellar crystals, higher-order structures⁹, chain conformations, and unit-cell packing¹⁰ – significantly influence the optical, thermal, and mechanical properties.

Page 17, Lines 282–284

Before: Precise control of the structure and properties of semicrystalline polymers will reduce the weight and use of polymeric materials, diminishing fossil fuel utilization as well as marine microplastic production, thereby helping realize a sustainable society.

After: Precise control of the structure and properties of semicrystalline polymers will **curtail the weight and utilisation** of polymeric materials.

(a) The description of the analysis of chain tilt from the microdiffraction data is not detailed enough. For instance, it leaves me wondering if a tilt contribution by chains out of the plane of the image has been overlooked. In Figure 4, particularly parts (b) and (d), the authors seem to make the assumption that the chains throughout both lamellae lie exactly in the plane of the paper. This would seem to be a favorable coincidence that needs to be remarked upon. In the supporting information, the authors do write that “ultrathin specimens for transmission electron microscopy (TEM) observation were carefully prepared so that the chain axis (c-axis) is normal to the electron beam.” This procedure needs to be explained in the main text, and the spread of angles from the normal (90 degree) orientation needs to be estimated. Is it really true that all chain axes in Figure 3 and Fig. 9, in all the different lamellae and with the indicated wide range of different chain tilt angles, were exactly perpendicular to the electron beam?? If the chains were tilted relative to the plane of the paper, the visible tilt would be smaller than the actual tilt, which would reduce the discrepancy with the literature. One needs to consider what happens to chains that were not parallel to the sample surface, when they reach that surface. Are they cut or forced to lay flat?

→ Thank you for pointing out the lack of a description for the chain-tilting angles with

respect to the plane of the image. In our study, we exclusively analysed lamellae with chains that were approximately oriented along the in-plane direction of the image (parallel to the plane of the image). In the following section, we have provided the reasons for assuming that the chains in the analysed lamellae were parallel to the plane of the image.

Page 7, Lines 113–140 (Supplementary Information)

Relationship between the electron beam direction and c-axis

The relationship between the electron beam direction and c-axis was established using the following four ED pattern rules:

- (i) When a hexagonal-like ED pattern is obtained (for example, Fig. 2e), the electron beam was parallel to the c-axis. (The c-axis was oriented perpendicular to the plane of the image).
- (ii) When 002 spots were observed, the electron beam was perpendicular to the c-axis.
- (iii) When two $hk0$ spots were observed, the angle between the electron beam and c-axis (molecular chain) could not be determined.
- (iv) When the flat faces of the lamellar crystals were $\{h0l\}$ planes¹⁻⁴, the lamellae viewed along the b -axis appeared to be edge-on. In this case, the c-axis was perpendicular to the electron beam (and parallel to the plane of the image), and a pair of 200 spots were observed in the ED patterns (Supplementary Fig. 8).

When the lamellar crystals were viewed edge-on, an image of the lamellae with clear contrast was acquired. However, as the lamellae inclined against the observing (electron-beam) direction, the apparent lamellar thickness (l_c') increased and the contrast became ambiguous. The relationship between the tilting angle of the lamellae (ψ) and l_c' is shown in Supplementary Fig. 7, which was constructed using the lamellar thickness estimated by SAXS ($l_c = 18$ nm). When ψ reached 7° , l_c' equalled the long spacing value (30.3 nm; $l_c +$ amorphous layer thickness), and the lamellar and amorphous domains merged into a single region. Therefore, when the ~ 18 -nm-thick lamellae were visualised edge-on (showing clear boundaries between the lamellar and amorphous domains), the tilting angles of the lamellae against the electron-beam direction were less than 7° . Additionally, when the lamellae reconstructed from the 200 spots were visualised edge-on, the electron beam was almost perpendicular to the c-axis ($90 \pm \sim 7^\circ$).

Supplementary Fig. 7 (shown below) was included in the SI file.

Supplementary Fig. 7 | a, Schematic depicting the increase in the apparent lamellar thickness (l'_c) due to the tilt of the lamellar crystals. **b**, Relationship between the tilt angle of the lamellar crystals (ψ) and l'_c . The dotted line indicates the long spacing (30.3 nm; lamellar thickness + amorphous layer thickness); lamellar crystals were not observed in the reconstructed image when l'_c was higher than this value.

Because the relationship between the electron beam and c-axis direction could not be determined exclusively from the ED patterns (lamellar morphologies had to be considered simultaneously), we have deleted the misleading description provided on Page 10, Lines 156–158. (Part of the description has been revised in response to other comments).

Before: Therefore, the two-spot $hk0$ peaks symmetric about the beam centre (Figs. 2c,d) indicate that the incident electron beam was perpendicular to the chain axis, with the chain orientation being vertical to the two-peak-connecting line.

After: Therefore, the two-spot $hk0$ peaks symmetric about the beam centre (Figs. 2c,d) indicated that the chain orientation was perpendicular to the line connecting the two peaks.

The following description of the relationship between the electron beam and c-axis has been added to the section on Page 11, Lines 182–187:

Because the lamellar crystals shown in Fig. 3a were reconstructed using the intensity of the 200 spots and viewed edge-on (with most lamellae having thicknesses similar to that obtained by SAXS, that is, ~18 nm), the chains in the lamellae were oriented along the in-plane direction of the image (see Supplementary Information and Supplementary Figs. 7 and 8 for more details).

The sentence highlighted by the reviewer – “...ultrathin specimens for transmission electron microscopy (TEM) observation were carefully prepared so that the chain axis (c-axis) is normal to the electron beam.” was modified as follows because it was misleading:

Page 2, Lines 15–20 (Supplementary Information)

Before: First, ultrathin specimens for transmission electron microscopy (TEM) observation were carefully prepared so that the chain axis (c-axis) is normal to the electron beam.

After: **In these studies**, ultrathin specimens for transmission electron microscopy (TEM) observation were prepared **initially to ensure** that the chain axis (c-axis) was normal to the electron beam; **this was confirmed by examining the morphologies of spherulites and lamellar crystals (for example, via TEM observations of stained and etched specimens) and the average molecular chain orientation (for example, via X-ray diffractometry).**

(b) As the authors know (line 67), at the end of lamellae the driving force for chain tilt is reduced. It seems that their analysis implicitly selects the ends of lamellae at the surface of the film. Just like the reconstructed surface of a crystal is different from the bulk, the lamellae near the surface here may have a different tilt than those in the bulk.

→ We agree with the points raised by the reviewer here. The lamellae near the surface may have a different tilt than those in the bulk; however, we did not select the ends of lamellae at the film's surface. For our experiments, we prepared the section *from the inner part of a bulk specimen* with various lamellar orientations using an ultramicrotome (Supplementary Fig. 10). We searched and selected an appropriate field of view for evaluating the chain tilt (that is, most of the lamellae were nearly edge-on against the incident electron beam). The sample preparation details were added to the Methods section.

Page 19, Lines 317–319

Before: An EM UC7 microtome (Leica Microsystems) was used to prepare ultrathin (~100 nm) sections of the crystallized films, which were produced under cryo-conditions (−155 °C) and collected on carbon-mesh microgrids with pore diameters of a few micrometres.

After: An EM UC7 microtome (Leica Microsystems) was used to prepare ultrathin (~100 nm) sections **along the film thickness direction** under cryo-conditions (−155 °C), which were then collected on carbon-mesh microgrids with pore diameters of a few micrometres.

In addition to these revisions, we have included Supplementary Fig. 10 in the SI file.

Supplementary Fig.10 | Schematic describing the preparation of ultrathin sections. The central part (b) of a crystallised film (a) was cut off, and ultrathin sections parallel to the sides of the specimen were prepared (c).

(c) The initial framing around the claim (line 20) that the tilt angle values “have been disputed for decades” does not seem quite appropriate. There is actually a general consensus in the literature, which is later acknowledged (in line 232, referring to the “commonly accepted value of 35 degrees”). It is the smaller values reported here that may stir up a dispute.

→ We appreciate the reviewer for providing this valuable comment. We apologise for providing inconsistent descriptions of the chain tilt in our manuscript. We agree with the consensus that the thermodynamically preferred chain-tilting angle of high-molecular-weight PE is ~35°. Moreover, as mentioned by the reviewer, it is not strange that the chain-tilting angle measured in this study (15°) differed from the commonly accepted value (35°) because we examined the lamellar crystals in HDPE prepared by short-duration annealing. In the revised manuscript, we have added a discussion on the reasons for our PE specimen exhibiting a chain tilt of 15°. However, because the revised part (pages 13–21) is too long to include in this letter,

we have only shown the revised figures (Figs. 3 and 4) below. Please see the revised text on the corresponding pages. Additionally, we have removed the descriptions in Lines 20 and 232.

Fig. 3 | **a**, Reconstructed dark-field (DF)-STEM image of HDPE. White triangles mark the locations from which the ED patterns shown in Figs. 2c–f were extracted, whereas red frames denote the analysed regions. **b**, Map of chain orientation angles in the Cartesian coordinate system (absolute molecular chain orientations). **c**, Straight and **d**, curved lamellae with a uniform chain orientation extracted from the left and central parts of Region 7 in **(b)**, respectively. **e**, Straight lamella with gradually changing chain orientations extracted from the upper-right part of Region 7 in **(b)**. **f**, Lamellae with two different chain orientations extracted from the central part of Region 5 in **(b)**. **g**, Lamellae corresponding to Region 2 in **(b)**, of which 1, 2, and 3 are non-parallel, whereas 4 is stacked.

Fig. 4 | a, Map of chain-tilting angle ϕ with respect to Figs. 3a and b. **b**, Histogram of ϕ for all analysed lamellae. The broad peak exhibits a peak top at $\sim 15^\circ$. **c,d**, Stacked lamellae from Regions 1 and 5, respectively, with small ϕ values ($< 15^\circ$). **e,f**, Isolated lamellae from Regions 2 and 7, respectively, with large ϕ values (close to 35°).

Clarifications needed:

(d) In the figure captions, “of PE” needs to be replaced with a more detailed brief description of the PE. The approximate number-average molecular weight needs to be specified – as the authors have implied, e.g. in line 67, at lower MW, with more chain ends, the small tilt angles found here would be less surprising. Similarly in the text, in line 156, these characteristics of “a PE film” need to be specified. What is the degree of branching?

→ We thank the reviewer for providing helpful suggestions. Although it is difficult to directly estimate the molecular weight from the melt flow rate (MFR), the extremely low MFR of our sample (0.06 g/10 min) suggests a fairly high molecular weight (probably $\geq 10^5$ g/mol⁴⁸). Therefore, the density of the chain ends decreased, and the chain-tilting angle was expected to increase. In the

revised manuscript, “polyethylene (PE)” was changed to “high-density polyethylene (HDPE)” based on the information provided in the product catalogue. In fact, the characteristic morphological features of HDPE with minimal branching (high crystallinity and few lamellar branches) were observed in the stained TEM image (Supplementary Fig. 4) and the reconstructed image (Fig. 3a) of our sample. Moreover, we changed “PE film” to “pressed HDPE film”. We have made the following revisions to the manuscript in this regard:

We have added another pertinent reference.

48. Teresa Rodríguez-Hernández, M., Angulo-Sánchez, J. L. & Pérez-Chantaco, A. Determination of the molecular characteristics of commercial polyethylenes with different architectures and the relation with the melt flow index. *J. Appl. Polym. Sci.* **104**, 1572–1578 (2007).

Page 17, Lines 288–297

Before: PE pellets (Novatec HD HF310; Nippon Polychem; density, 0.950 g/cm³; melt flow rate, 0.06 g/10 min) were processed into 0.5-mm-thick films by hot pressing. The pressed films were melted at 160 °C for 5 min and further annealed at 120 °C for 1 h for crystallization. The lamellar crystals in the film were assumed to be randomly oriented.

After: HDPE pellets (Novatec HD HF310; Nippon Polychem; density = 0.950 g/cm³; melt flow rate = 0.06 g/10 min; **weight averaged molecular weight (M_w) $\geq 10^5$ g/mol⁴⁸**) were processed into a 0.5-mm-thick film by hot pressing. **The pressed film was cut into a 20 mm square and then melted on a hot plate at 160 °C for 5 min. The melted film was then quickly moved to another hot plate set to 120 °C and maintained for 1 h to permit crystallisation. After the thermal annealing, the film was rapidly cooled on a metal surface precooled with liquid nitrogen to freeze the structure. Approximately 1 mm of the outer edge of the film was cut off, and the central part was used as the sample in a series of measurements. Therefore,** the lamellar crystals in the sample were assumed to be randomly oriented.

(e) In the caption of Figures 2 c-e, it is stated that “the direction of the incident electron beam is indicated by [uvw]” but numbers in square brackets are shown only in (e). What is [uvw] for (c) and (d)? And could the locations of c, d, and e please be marked in (b), and in Figure 3?

→ We thank the reviewer for these comments. We deliberately did not use [uvw] in Figs. 2c and d because the crystal orientations could not be determined using only one pair of diffraction

peaks. The locations from which the ED patterns were extracted have been added to Figs. 2b and 3. We have revised the captions of these figures as follows:

Page 32, Lines 543–547

Before: **c–e**, Indexing of representative ED patterns acquired via the NDI of PE. The direction of the incident electron beam is indicated by $[uvw]$, where u , v , and w are the smallest integers with no common divisor.

After: **c–e**, Indexing representative electron diffraction (ED) patterns acquired via the NDI of HDPE, which were extracted from the tips of the white triangles shown in **b**. In **(e)**, the direction of the incident electron beam is indicated by $[uvw]$, where u , v , and w are the smallest integers with no common divisor.

Page 32, Lines 551–553

Before: **Fig. 3 |** Reconstructed DF-STEM image of PE. The magenta and red frames highlight the analysed regions.

After: **Fig. 3 | a**, Reconstructed dark-field (DF)-STEM image of HDPE. White triangles mark the locations from which the ED patterns shown in Figs. 2c–f were extracted, whereas red frames denote the analysed regions.

(f) Line 180: “chain orientation being vertical to the two-peak connecting line.” What does “vertical to” mean? Please rephrase for more clarity.

→ We thank the reviewer for pointing out this grammatical mistake. The “vertical to” was changed to “perpendicular to” as follows:

Page 10, Lines 156–158

Before: Therefore, the two-spot $hk0$ peaks symmetric about the beam centre (Figs. 2c,d) indicate that the incident electron beam was perpendicular to the chain axis, with the chain orientation being vertical to the two-peak-connecting line.

After: Therefore, the two-spot $hk0$ peaks symmetric about the beam centre (Figs. 2c,d) indicated that the chain orientation was perpendicular to the line connecting the two peaks.

(g) Line 213: “chain tilt from the perpendicular c-axis orientation” also is unclear: the chain is along

the c-axis, so how can there be “tilt from” it? This needs to be rephrased for clarity.

→ We thank the reviewer for this helpful advice. We agree that the chain axis should be parallel to the c-axis. To that end, we have removed the sentence on Line 213 and have heavily revised the discussion part.

(h) The title “Reassessing chain tilt in the lamellar crystals of semicrystalline polymers” seems too general for a study of one sample of polyethylene. “Reassessing chain tilt in the lamellar crystals of semicrystalline polyethylene” would be more accurate.

→ We have modified the title based on the reviewer’s suggestion.

Title:

Before: Reassessing chain tilt in the lamellar crystals of semicrystalline polymers

After: Reassessing chain tilt in the lamellar crystals of polyethylene

Replies to Reviewer #3

This communication reports the application of a fairly sophisticated scanning transmission electron microscopy (STEM) technique dubbed “nanodiffraction imaging” (NDI) to study the tilt of chain stems within lamellar crystals of polyethylene (PE) crystallized from the melt. The new thing is apparently the very fine beam of electrons that allows a lateral resolution as small as 1-2 nm, much finer than conventional beams of X-rays, electrons or neutrons, which are on the order of micrometers. Using this high resolution beam, the authors report the analysis of electron diffraction patterns from single lamellae of polyethylene in ultrathin sections (100 nm thick) obtained by cryomicrotomy of melt-crystallized thin films.

The technique is valuable and its application to the examination of tilt angles in PE is interesting. Other aspects of the presentation, however, are bothersome from the polymer science perspective, and tend to overstate the significance of the result. A few of these aspects are highlighted below.

→ We appreciate the reviewer's careful reading and their valuable comments, which have helped considerably in strengthening our manuscript. In the following sections, we have responded to the reviewer's questions/comments. Please note that the modifications in the revised manuscript have been highlighted in red.

(1) The authors seem to imply incorrectly that tilt angle in semicrystalline polymers is an inherent feature, without regard to chemistry or sample history. Many semicrystalline polymers do not exhibit a clear-cut lamellar morphology at all. Both the title and the opening paragraph should be revised to mention PE, the polymer studied in this work, explicitly.

→ We greatly appreciate the reviewer's suggestions and agree with the points raised above. Therefore, we have revised the title and opening paragraph to explicitly mention PE, as follows:

Title:

Before: Reassessing chain tilt in the lamellar crystals of semicrystalline polymers

After: Reassessing chain tilt in the lamellar crystals of polyethylene

Page 2, Lines 14–16

Before: In 1957, Keller deduced that long polymer chains fold to form thin lamellar crystals, with

the molecular chains being perpendicular to the flat face of the crystals (the chain-folding model)¹.

After: In 1957, Keller deduced that long **polyethylene (PE)** chains evidently fold to form thin single lamellar crystals, with the molecular chains perpendicular to the flat face of the crystals (the chain-folding model).

(2) Even in those polymers, like PE, that do exhibit lamellar crystal morphologies, the lamellar morphology is highly dependent on the source of PE (how synthesized, molecular weight, etc.) and how the sample was crystallized (whether from solution or melt, at what temperature, cooling rate or thermal history, and whether flow was involved). Thus, it is not necessarily surprising that samples of semicrystalline polyethylene prepared in different labs under different conditions exhibit different tilt angles.

→ We thank the reviewer for this comment. Although $\sim 35^\circ$ may be a thermodynamically stable tilting angle, we agree that PE specimens with different primary structures crystallised under other conditions can exhibit different angles. Because of the reviewers' comments, we have reconsidered the novelty of our paper. Consequently, we have redirected the focus of our paper from determining the major chain tilt in the specimen to (i) visualising the nanoscale distribution of both lamellar crystals and chain orientations, and (ii) clarifying the formation mechanisms of lamellar crystals at each location. To reflect these points, we have made significant changes to our manuscript, as follows:

Page 2, Lines 21–28

Before: Here we report the direct determination of molecular chain orientations in lamellar crystals using a novel electron-diffraction-based imaging technique with nanometre-scale positional resolution¹⁴ and provide compelling evidence for the existence of diverse φ values (primarily $\sim 15^\circ$). Greater clarification of the nanoscale structure–property relationships of semicrystalline polymers can permit precise tuning of their properties and pave the way for lightweight resource-saving material design.

After: Here, we report the direct determination of molecular chain orientations in the lamellar crystals **of high-density PE** using a novel electron-diffraction-based imaging technique with nanometre-scale positional resolution and provide compelling evidence for the existence of **lamellar crystals with different inner-chain orientations. Factors dictating the φ values of lamellae were determined.** Clarifying the nanoscale **variation in lamellar crystals and their formation mechanisms** in PE can permit precise tuning of properties

and expedite resource-saving material design.

Page 8, Lines 121–128

Before: In this study, the nanoscale chain orientation in lamellar crystals was “directly” observed by NDI. Chain tilt was experimentally evaluated without assumptions by analysing numerous position-resolved ED patterns. This precise chain-tilt analysis resolves the 40-year-old dilemma of linking the structural nature of polymer crystals to their physical properties.

After: In this study, the nanoscale **morphologies of lamellar crystals and the molecular chain orientations in lamellae were visualised simultaneously in a ‘direct’ manner using position-resolved ED patterns. Chain tilt in lamellae was experimentally identified without any assumptions through orientational relationships between the lamellae and their inner-chain orientations. Precise analysis revealed the variation in the lamellar morphology and the chain tilt inside lamellae for an HDPE specimen prepared by short-duration annealing. Furthermore, mechanisms governing chain tilt in lamellae are proposed.**

In addition to these corrections, we have significantly revised the discussion part of our manuscript. Because those revisions are too extensive to include in this letter, please see Pages **11–17** in the revised manuscript (changes are shown in red). In addition to the corrections made to the discussion part, we have modified Figs. 3 and 4, as follows:

Fig. 3 | a, Reconstructed dark-field (DF)-STEM image of HDPE. White triangles mark the locations from which the ED patterns shown in Figs. 2c-f were extracted, whereas red frames denote the analysed regions. **b**, Map of chain orientation angles in the Cartesian coordinate system (absolute molecular chain orientations). **c**, Straight and **d**, curved lamellae with a uniform chain orientation extracted from the left and central parts of Region 7 in **(b)**, respectively. **e**, Straight lamella with gradually changing chain orientations extracted from the upper-right part of Region 7 in **(b)**. **f**, Lamellae with two different chain orientations extracted from the central part of Region 5 in **(b)**. **g**, Lamellae corresponding to Region 2 in **(b)**, of which 1, 2, and 3 are non-parallel, whereas 4 is stacked.

Fig. 4 | a, Map of chain-tilting angle ϕ with respect to Figs. 3a and b. **b**, Histogram of ϕ for all analysed lamellae. The broad peak exhibits a peak top at $\sim 15^\circ$. **c,d**, Stacked lamellae from Regions 1 and 5, respectively, with small ϕ values ($< 15^\circ$). **e,f**, Isolated lamellae from Regions 2 and 7, respectively, with large ϕ values (close to 35°).

(3) Recognizing that tilt angle is sensitive to crystallization conditions, it is somewhat of an overstatement to represent discrepancies in reported values as some great dilemma. Unless all aspects of sample chemistry and history are controlled, the reporting of different tilt angles by different labs (such as refs 8,9 and ref 10) is revealing, but neither is necessarily “wrong”. Even in this work, where the peak values are close to 5 or 15 deg, the range of tilt angles reported is quite large (up to 70 deg). Thus, assertions like (p15) “the predominant phi value in the specimen was experimentally proven to be 15° herein, which differs from the commonly accepted value of 35° ” come across as misleading and argumentative. After all, how can the value of 35° be both “commonly accepted” and “disputed for decades” (p2)?

→ We thank the reviewer for these queries and agree with the points raised above. We did not intend to claim that the peak of the chain-tilting angle distribution ($\phi = 15^\circ$) was a new general

value for PE. As mentioned above, the main findings of this study were that the chain tilt varies with the lamellar crystal and location in an HDPE specimen, which can elucidate the formation mechanisms of the lamellar crystals. We have removed the misleading sentences and have significantly revised the discussion part in the revised manuscript. Please see the portions modified based on the reviewer's comment (2).

(4) There is one value of tilt angle that might be special: the thermodynamically most favored one. The discussion of the increase of tilt angle with increasing T_c (p15,16), reported by ref 10 and also in this work, is consistent with a larger tilt angle being thermodynamically favored, as also argued in ref 13. It suggests that the "proven" value of 15 deg reported here is probably a consequence of the prevailing crystallization kinetics during sample preparation. At the very least, the authors should provide sufficient detail about the PE used, film preparation and thermal history in the main text or SI that the crystallization conditions could be reproduced. The single sentence provided on p10 is not enough.

→ We thank the reviewer for providing valuable advice. In the revised manuscript, we have added the following discussion on the factors determining the chain tilt in certain types of lamellar crystals:

Page 14, Lines 234–260

The reasons for the stacked and isolated lamellar crystals exhibiting different chain-tilting angles are now discussed. As described above, amorphous chains tend to be arranged near the flat faces of the lamellae to abate the density anomaly, considering the three effects of chain ends, sharp folds, and chain tilt¹⁴. The impact of chain ends on the flat faces of the lamellae could be ignored¹⁴ for our specimen because of its high molecular weight (weight averaged molecular weight $M_w \geq 10^5$ g/mol, estimated from the extremely low melt flow rate of 0.06 g/10 min⁴⁸). Thus, the density of sharp folds near the flat faces of the lamellae was presumably the only factor affecting the chain tilt in the lamellae. The density anomaly near the lamellar surfaces could be mitigated at low sharp fold densities. Therefore, in this case, the chains in the lamellae could reduce the chain-tilting angles.

In our specimen, the stacked lamellae exhibited smaller chain-tilting angles ($\varphi < 15^\circ$) than that of the isolated lamellae ($\varphi \sim 35^\circ$). This discrepancy indicated that the density of sharp folds near the flat faces of the stacked lamellae was lower than that of the isolated lamellae. The formation mechanism of the stacked lamellae can thus be discussed. During the growth of the stacked lamellae, long chains emanating from the first lamellae are incorporated into the lamellae growing next to the first lamellae, resulting in tie chains between the neighbouring lamellae⁴⁷. Owing to the presence of

the tie chains (interlamellar interactions), the stacked lamellae likely exhibited a smaller sharp fold content near the flat faces than that in the isolated lamellae. Thus, smaller chain-tilting angles were favoured for the stacked lamellae. The numerous stacked lamellae with small chain-tilting angles in the presently analysed specimen led to the histogram of φ (Fig. 4b) exhibiting a peak top at $\sim 15^\circ$. Local analysis indicated that two types of lamellar crystals (stacked and isolated) were present in a single specimen, and that they exhibited different chain-tilting angles with different formation mechanisms. Furthermore, a relationship was established between the factors affecting the density of sharp folds near the flat faces of the lamellae and the chain tilt inside the lamellae.

Page 16, Lines 269–277

Furthermore, interlamellar interactions were found to be a key factor in determining the chain-tilting angles inside the lamellae; while isolated lamellae exhibited φ values close to the thermodynamically favoured angle (35°), stacked lamellae preferred φ values smaller than 35° owing to the decrease in the sharp fold density caused by the tie chains between neighbouring lamellae. It is worth noting that the observations in this study (particularly the histogram of φ) depend on the primary structure and thermal history of the analysed specimen and can vary with the crystallisation conditions. For example, well-annealed HDPE specimens have shown roof-like lamellae with only one chain-tilting angle of $\sim 35^\circ$ ^{22,23}.

Additionally, we have added the following details concerning the sample preparation to the Methods section:

Page 17, Lines 288–297

Before: PE pellets (Novatec HD HF310; Nippon Polychem; density, 0.950 g/cm³; melt flow rate, 0.06 g/10 min) were processed into 0.5-mm-thick films by hot pressing. The pressed films were melted at 160 °C for 5 min and further annealed at 120 °C for 1 h for crystallization. The lamellar crystals in the film were assumed to be randomly oriented.

After: HDPE pellets (Novatec HD HF310; Nippon Polychem; density = 0.950 g/cm³; melt flow rate = 0.06 g/10 min; weight averaged molecular weight (M_w) $\geq 10^5$ g/mol⁴⁸) were processed into a 0.5-mm-thick film by hot pressing. The pressed film was cut into a 20 mm square and then melted on a hot plate at 160 °C for 5 min. The melted film was then quickly moved to another hot plate set to 120 °C and maintained for 1 h to permit crystallisation. After the thermal annealing, the film was rapidly cooled on a metal surface precooled with liquid nitrogen to freeze the structure. Approximately 1 mm of the outer edge of the film was cut off, and the central part was used as the sample in a series of measurements. Therefore, the lamellar crystals in the sample were assumed to be randomly oriented.

(5) On p9, the authors assert that their “precise chain-tilt analysis resolves the 40-year-old dilemma of linking the structural nature of polymer crystals to their physical properties.” If referring to optical, thermal or mechanical properties mentioned in p3, what is the new “linkage” to those properties that NDI provides? There is no real debate in the polymer community that properties depend on structure, but how those properties depend specifically on tilt angle could be better articulated. The assertion that the current work will somehow have measurable impact on marine microplastics seems particularly tenuous.

→ We greatly appreciate the reviewer’s comments, which have helped us refine our manuscript. In this context, we have described the variation in lamellar morphology and the chain tilt inside lamellae in HDPE prepared by short-duration annealing in the revised manuscript. Furthermore, we have proposed the determinant mechanisms of chain tilt in the lamellae. Elucidating the chain-tilt mechanisms could permit fine-tuning of the internal structures and, consequently, the physical properties of PE specimens. We have added the following sentences in the main text in this regard:

Page 17, Lines 277–280

Because the stacked and isolated lamellar crystals would exhibit different physical properties owing to their differing chain-tilting angles, the properties of PE could be readily tweaked by controlling the amount and ratio of these lamellae in an appropriate preparation method.

Other reviewers have also pointed out the inadequate discussion of environmental issues in our original manuscript. We agree that the logic underlying these claims must be substantiated. The statement was included in the original manuscript because we thought that the thermodynamic stability of lamellar crystals, which is heavily influenced by the molecular chain tilt, is related to the decomposition of marine microplastics. Amorphous chains are believed to preferentially degrade during seawater- and sunlight-induced decomposition owing to an increase in crystallinity. The subsequent step of the degradation is the decomposition of the crystalline region. Generally, crystals with greater considerable thermodynamic stability are less influenced by external stimuli such as heat and other forces. Hence, we presumed that more thermodynamically stable crystals would also be less affected by seawater and sunlight. Essentially, the decomposition of lamellar crystals could be controlled by tuning the chain tilt, thereby lowering the environmental impact of polymeric materials.

However, this significant leap in logic could not be explained adequately in the manuscript.

Thanks to the reviewers' comments, we recognised that the thermophysical and mechanical properties deserved greater attention than the impact on marine microplastics. Therefore, we have deleted the description of marine microplastics and included the following description:

Page 3, Lines 41–45

Before: Thus, the crystallization-dependent hierarchical structures of semicrystalline polymers (Extended Data Fig. 1) significantly affect their physical properties.

The fraction, size, and orientation of hierarchical structural elements, including lamellar crystals, higher-order structures²¹, chain conformation, and unit-cell packing²², significantly influence optical, thermal, and mechanical properties. For example, the enzymatic hydrolysis of crystallized chains is considerably slower than that of amorphous chains in spherulite-containing polyesters²³. These insights help elucidate the formation mechanism of marine microplastics²⁴. PE is the most mass-produced and mass-discarded plastic, and is predominantly responsible for marine microplastic pollution²⁵. Therefore, insights into the internal nanostructures of PE and their influence on the marine microplastic formation mechanism can alleviate polymer-related environmental issues.

After: Thus, the crystallisation-dependent hierarchical structures (Supplementary Fig. 1) significantly affect the physical properties of the polymers. The fraction, size, and orientation of the hierarchical structural elements – including lamellar crystals, higher-order structures⁹, chain conformations, and unit-cell packing¹⁰ – significantly influence the optical, thermal, and mechanical properties.

Page 17, Lines 282–284

Before: Precise control of the structure and properties of semicrystalline polymers will reduce the weight and use of polymeric materials, diminishing fossil fuel utilization as well as marine microplastic production, thereby helping realize a sustainable society.

After: Precise control of the structure and properties of semicrystalline polymers will **curtail the weight and utilisation** of polymeric materials.

In addition, clarification of the following points is recommended.

(6) Assuming that the diffraction patterns are collected in transmission mode (Fig 2), the resolution in the thickness direction would seem to be controlled by the sample thickness (~100 nm), which is much larger than the lamellar thickness but smaller, perhaps, than the lamellar width. If the latter is not the case, then I would expect the ED pattern to comprise multiple lamellae (like Fig

2f). Some mention in the main text about the sample thickness used and the restriction imposed by the condition on lamellar width is warranted.

→ We thank the reviewer for providing this helpful comment. As indicated by the reviewer, the resolution in the thickness direction was controlled by the sample thickness (~100 nm), which was considerably greater than the lamellar thickness (~20 nm) but smaller than the lamellar width (probably hundreds of nm, as estimated from the lamellar morphologies observed in the staining-TEM and reconstructed DF-STEM images). Therefore, the ED pattern was likely derived from a single lamellar crystal with a nearly edge-on geometry.

In certain positions, an overlap of the edge-on lamellar crystals was suggested by the four-spot ED patterns and the morphology shown in the reconstructed image. These locations were omitted from the chain-tilting angle analysis. We have added the following sentence to the revised manuscript in this regard:

Page 9, Lines 136–141

Before: Certain ultrathin (~100 nm) sections were *stained* with RuO₄ for TEM (Extended Data Fig. 4), which revealed stacked-lamellar structures with an undulating stripe morphology.

After: Certain ultrathin (~100 nm **thick**) sections were *stained* with RuO₄ for TEM **analysis** (Supplementary Fig. 4), which revealed stacked-lamellar structures with an undulating stripe morphology. **Because the lengths of most lamellar crystals in their in-plane direction were greater than the thickness of the ultrathin section, the edge-on lamellar crystals were considered to penetrate the section.**

(7) P12: The phrase “chain orientation being vertical to the two-peak-connecting line” is unclear. Do the authors mean “perpendicular”? Also, “line connecting the two peaks” would be better than “two-peak-connecting line”.

→ We appreciate the reviewer’s advice. We have made the following corrections to the manuscript:

Page 10, Lines 156–158

Before: Therefore, the two-spot *hk0* peaks symmetric about the beam centre (Figs. 2c,d) indicate that the incident electron beam was perpendicular to the chain axis, with the chain orientation being vertical to the two-peak-connecting line.

After: Therefore, the two-spot *hk0* peaks symmetric about the beam centre (Figs. 2c,d)

indicated that the chain orientation was perpendicular to the line connecting the two peaks.

(8) All of the Extended Data figures appear to be referenced by the wrong figure numbers in the SI. It might also be the case for Extended Data figures referenced in the main text. This mistake is a significant hindrance to reviewing the paper. Authors should check the figure numbering carefully.

→ We apologise for these rudimentary mistakes. We have corrected the referenced numbers of Supplementary figures throughout the manuscript. Moreover, to comply with the journal guidelines, we have changed the Extended Data figures to Supplementary Figures and have moved them to the SI file.

(9) Extended Data Fig 8 caption refers to “parallel and antiparallel chains of the orthorhombic PE.” However, unlike polypropylene, the PE chains do not have a unique direction; the parallel/antiparallel distinction is meaningless. (It is the setting angle that distinguishes the two chains in the unit cell.)

→ We thank the reviewer for pointing out the inappropriate description. We have revised the description of the two types of HDPE molecular chains in the captions of Fig. 1a and Supplementary Fig. 8.

Page 31, Lines 527–530

Before: Two planar-zigzag-type PE chains occupy the unit cell, with the light-green-coloured central chain oriented opposite to the dark-green-coloured corner ones.

After: Two types of planar-zigzag-structured PE chains – shown as a light-green central chain and dark-green corner ones – occupy the unit cell with different setting angles (the angle between the *bc* and zigzag planes).

Page 19, Lines 227–229 (Supplementary information)

Before: Light-green- and dark-green-coloured objects (filled squares and wavy lines) represent parallel and antiparallel chains of the orthorhombic PE, respectively.

After: Light-green and dark-green objects (filled squares and wavy lines) represent the two PE chains packed in the orthorhombic unit cell with different setting angles.

Reviewers' Comments:

Reviewer #1:

Remarks to the Author:

The authors have taken in account the many suggestions presented by the different reviewers. The focus of the paper has been modified.

The paper is suitable for publication. It remains that this elaborate technique has been applied here to a relatively marginal problem in polymer science. The very fact that chain tilts are so diverse in polyethylene indicates that there is no simple or single explanation to offer for this variety. In other words: the technical demonstration is made. Persuing this line may not be sufficiently rewarding.

Reviewer #2:

Remarks to the Author:

The authors have engaged with the reviewers' comments in a very constructive manner and made most of the suggested changes. They have removed overly broad claims and instead focused on making the technical description clearer, including specifics of the material studied. The observed difference between isolated and stacked lamellae is intriguing. Some of the arguments related to sharp fold densities on page 14-16 eluded me but that's okay. The figures are stunning and the text is well written, so I consider this work very suitable for Nature Communications.

Technical details:

The features in Figure 2 are very small (in particular compared to those in Figure 1). Maybe in Figure 2, panels (c) – (f) could be moved below panels (a) and (b).

Bottom of p. 11 (line 187): "parallel to THE b-axis"

Line 232/233: "the small chain-tilting angles observed in the stacked lamellae ... would also be preferred values for the lamellae." Of what? The end of the sentence baffled me, in particular since a few lines earlier lamellae with large chain-tilting angles of 31, 28, and 30 degrees were discussed.

Reviewer #3:

Remarks to the Author:

This is a thoroughly rewritten communication. The results have not changed significantly, but their interpretation has been altered dramatically. The focus is now on the diversity of relationships between inner-chain orientation and the lamellae within which these chains reside. It is a significant improvement over the original exposition, and eliminates many of the concerns raised in the original review.

Based on an unspecified, perhaps small, number of observations, the authors then associate small tilt angles with stacked lamellae and large tilt angles with isolated lamellae. This is despite the fact that the distribution of tilt angles in Fig 4b does not hint at the existence of two such distinct populations. However, if it can be shown to be statistically significant, this association would be significant and thought-provoking.

The real problem arises with the reasoning proposed for the selection of small tilt angles in stacked lamellae, which appears to be flawed. Referring back to the original arguments of Sir Charles Frank (Faraday Discuss. Chem. Soc. 1979, 68, 7), which are cited in ref 14, in the absence of significant contributions from chain ends, $(1-p)$ and $\cos(\theta)$ should be inversely related, where p is the fraction of folds and θ is the tilt angle. As tilt angle increases (i.e. $\cos(\theta)$ decreases), the

fraction of folds decreases (i.e. $(1-p)$ increases). That is, low fold density is mitigated by large tilt angle, in contrast to the argument presented by the authors on p15. The reasoning that tie chains form at the expense of folds in stacked lamellae then is not consistent with the observation of small tilt angles in stacked lamellae. Since the methods reported here do not image molecular features like folds or tie chains, the proposed formation mechanism cannot be readily confirmed or refuted. Publication of the manuscript with this newly introduced shortcoming is not recommended.

Reply to Reviewer #1

The authors have taken in account the many suggestions presented by the different reviewers. The focus of the paper has been modified.

The paper is suitable for publication. It remains that this elaborate technique has been applied here to a relatively marginal problem in polymer science. The very fact that chain tilts are so diverse in polyethylene indicates that there is no simple or single explanation to offer for this variety. In other words: the technical demonstration is made. Persuing this line may not be sufficiently rewarding.

→ We greatly appreciate reviewer #1's recommendation of our manuscript for publication. Although we tried to explain our experimental results in the previous revision, we agree that the chain tilts were diverse in our experiment, and no single explanation is possible. We need to design a more controlled experiment for a detailed discussion of the chain tilt, which we plan to do in future research.

According to the other reviewers' comments, we made changes to the manuscript, which are highlighted in **red**.

Replies to Reviewer #2

The authors have engaged with the reviewers' comments in a very constructive manner and made most of the suggested changes. They have removed overly broad claims and instead focused on making the technical description clearer, including specifics of the material studied. The observed difference between isolated and stacked lamellae is intriguing. Some of the arguments related to sharp fold densities on page 14-16 eluded me but that's okay. The figures are stunning and the text is well written, so I consider this work very suitable for Nature Communications.

→ We greatly appreciate your positive recommendation of our manuscript. We agree with reviewer #2 that the arguments related to sharp-fold densities on pages 14–16 are somewhat confusing, primarily due to our speculative discussion based on unobservable folds and tie chains. Please note that our method directly measures chain tilt but provides no information about folds and tie chains. Thus, we deleted the related speculative discussions about the formation mechanisms of lamellar crystals. The modifications are highlighted in **red** in the revised manuscript.

Pages 14–15, Lines 236–257

The reasons for the isolated and stacked lamellar crystals exhibiting different chain tilt angles are discussed. As described above, amorphous chains **are** arranged near the flat faces of the lamellae to **moderate** the density anomaly, considering the three effects of chain ends, sharp folds, and chain tilt¹⁴. Because our specimen had a high molecular weight (weight-averaged molecular weight $M_w \geq 10^5$ g/mol, estimated from the low melt flow rate of 0.06 g/10 min⁴⁸), the fraction of the chain ends in the specimen was small. The **moderation of** the density anomaly due to the chain ends was negligible¹⁴. Thus, the **fraction** of sharp folds near the flat faces of the lamellae was presumably the only factor affecting the chain tilt in the lamellae. **Theory⁴⁸ and simulations¹³ predicted that the sharp-fold fraction decreases with an increase in the tilt angle. Fritzsche et al. used the sharp-fold fraction of $\sim 1/3$ and the thermodynamically stable φ of 34° to moderate the density anomaly near lamellar surfaces¹⁴. Our experimental result, i.e., smaller φ values in stacked lamellae than in isolated ones, implies that the sharp-fold fractions may differ between stacked and isolated lamellae. This difference may be due to the existence of tie chains that bridge two or more neighbouring lamellae in the case of stacked lamellae⁴⁷.**

Nanoscale analysis revealed that two types of lamellar crystals (isolated and stacked) were present in a single specimen and that they exhibited different chain tilt angles (Fig. 4i). The reason for such different tilt angles remains an open question and is expected to be revealed by simulating the crystallisation dynamics near the lamellar surface with the knowledge of the directly measured tilt angles of all the lamellae examined in the present study.

Technical details:

The features in Figure 2 are very small (in particular compared to those in Figure 1). Maybe in Figure 2, panels (c) – (f) could be moved below panels (a) and (b).

→ Thank you for the comment. We have modified Fig. 2 as follows.

Fig. 2 | a, Schematic describing nanodiffraction imaging (NDI). **b**, Bright-field scanning transmission electron microscopy (STEM) image of high-density PE (HDPE) acquired after NDI. The dashed square indicates the NDI scan area. **c–e**, Indexing representative electron diffraction (ED) patterns acquired via the NDI of HDPE, which were extracted from the tips of the white triangles shown in **(b)**. In **(e)**, the direction of the incident electron beam is indicated by $[uvw]$, where u , v , and w are the smallest integers with no common divisor. **f**, ED pattern of a polycrystalline region. **g**, Average of all the ED patterns. **h**, ED pattern of the same sample acquired from another field of view using a 2.1- μm -diameter electron beam.

Bottom of p. 11 (line 187): "parallel to THE b-axis"

→ Thank you for the comment. We revised the corresponding part as follows.

Page 11, Lines 181

Because the growth direction of PE lamellae is parallel to **the *b*-axis**, the edge-on lamellae grow perpendicular to the in-plane direction of the image.

Line 232/233: "the small chain-tilting angles observed in the stacked lamellae ... would also be preferred values for the lamellae." Of what? The end of the sentence baffled me, in particular since a few lines earlier lamellae with large chain-tilting angles of 31, 28, and 30 degrees were discussed.

→ Thank you for the comment. We revised the corresponding part as follows.

Page 14, Lines 226 and 227

These small **tilt** angles would also be preferred values for the **stacked lamellae in the present specimen**.

Replies to Reviewer #3

This is a thoroughly rewritten communication. The results have not changed significantly, but their interpretation has been altered dramatically. The focus is now on the diversity of relationships between inner-chain orientation and the lamellae within which these chains reside. It is a significant improvement over the original exposition, and eliminates many of the concerns raised in the original review.

→ We greatly appreciate your positive comments on our revisions. Please note that the modifications are highlighted in red in the revised manuscript.

Based on an unspecified, perhaps small, number of observations, the authors then associate small tilt angles with stacked lamellae and large tilt angles with isolated lamellae. This is despite the fact that the distribution of tilt angles in Fig 4b does not hint at the existence of two such distinct populations. However, if it can be shown to be statistically significant, this association would be significant and thought-provoking.

→ We thank the reviewer for his/her comment. We agree that it is difficult to see two distinct populations in the histogram in Fig. 4b. Because the number of isolated lamellae (chain tilt angle, $\varphi \approx 34^\circ$) is far smaller than the number of stacked lamellae ($\varphi \approx 15^\circ$), the peak for the isolated lamellae ($\varphi \approx 34^\circ$) did not appear clearly in the histogram. To demonstrate the different φ for stacked and isolated lamellae, we added the histograms of φ for the isolated and stacked lamellae separately (Fig. 4i). Furthermore, to make the data more statistically reliable, we analysed φ in two additional fields of view (Supplementary Figs. 12–14). The peak positions of stacked lamellae were clearly lower than those of isolated lamellae (Supplementary Fig. 14).

Pages 13 and 14, Lines 212–235

Next, the origins of the variation in φ were considered. Figs. 4c–e show maps of the chain tilt angle for isolated lamellae, with no parallel lamellae within 30 nm, extracted from Regions 2, 5, and 7 of Fig. 4a, respectively. The representative chain tilt angles in the lamellae in Fig. 4c, the upper part of the lamella in Fig. 4d, and the left and right lamellae in Fig. 4e were 31° , 25° , 28° , and 30° , respectively. The isolated lamellar crystals exhibited chain tilt angles close to the thermodynamically favoured value ($\varphi = 34^\circ$). These results indicated that the annealing condition used in the present study was sufficient to align the chains in isolated lamellae. The lower part of the lamella in Fig. 4d was considered a stacked lamellar region and exhibited an average chain tilt angle of 8° . Figs. 4f–h show maps of the chain tilt angle for the stacked lamellae extracted from Regions 1, 2, and 5 of Fig. 4a, respectively. In the case of Fig. 4f, the representative chain tilt angles in the upper and lower lamellae were 5° and 15° , respectively. Figs. 4g and 4h show similar tendencies (1° – 23°). Therefore,

the chain tilt angles for stacked lamellae were smaller than the thermodynamically stable value of 34° (Supplementary Figs. 12 and 13). These small tilt angles would also be preferred values for the stacked lamellae in the present specimen.

Figure 4i shows histograms of φ for the isolated and stacked lamellae. We note that all the lamellae shown in Fig. 4 and Supplementary Figs. 12 and 13 are included in the histograms of Figs. 4b and 4i. The histogram of the isolated lamellae exhibits a high frequency of tilt angles close to the thermodynamically stable value of 34° , while that of the stacked lamellae has peak positions unambiguously smaller than 34° . Because the ratio of the number of stacked lamellae to the total number of lamellae analysed was $\sim 94\%$, the integrated histogram of the isolated and stacked lamellae in Fig. 4b exhibited a single peak at $\sim 15^\circ$.

Fig. 4 | **a**, Map of the chain tilt angle φ with respect to Figs. 3a and b. **b**, Histogram of φ for all the lamellae shown in (a) and Supplementary Figs. 12 and 13. The histogram was normalized by the total area of the lamellae analysed for the determination of φ . The broad peak exhibits a peak top at $\sim 15^\circ$. **c,d,e**, Isolated lamellae from Regions 2, 5, and 7, respectively, with large φ values (close to 34°). **f,g,h**, Stacked lamellae from Regions 1, 2, and 5, respectively, with small φ values ($< 34^\circ$). **i**,

Histograms of φ for isolated and stacked lamellae. The histogram for isolated lamellae was normalized by the total area of the isolated lamellae, and that for stacked lamellae was normalized by the total area of the stacked lamellae. Because stacked lamellae were dominant in the present specimen, the histogram for all lamellae shown in (b) exhibits a feature similar to that for the stacked lamellae.

Supplementary Fig. 12 | a, DF-STEM image of HDPE reconstructed by summing the intensities of the 110, 200, 210, and 020 peaks. Note that this image has a different field of view (in the same ultrathin section) from the images in the main-text figures and Supplementary Figs. 11 and 13. The electron-beam scanning was performed at 5-nm intervals (6-nm intervals were used for Figs. 3 and

4). **b**, DF-STEM image of HDPE reconstructed by the 200 peak intensity. **c**, Map of the chain orientation angles in the Cartesian coordinate system (absolute molecular chain orientations). **d**, Map of the chain tilt angle φ .

Supplementary Fig. 13 | **a**, DF-STEM image of HDPE reconstructed by summing the intensities of the 110, 200, 210, and 020 peaks. This image has a different field of view (in the same ultrathin section) from the images in the main-text figures and Supplementary Figs. 11 and 12. The electron-beam scanning was performed at 5-nm intervals (6-nm intervals were used for Figs. 3 and 4). **b**, DF-STEM image of HDPE reconstructed by the 200 peak intensity. **c**, Map of the chain orientation

angles in the Cartesian coordinate system (absolute molecular chain orientations). **d**, Map of the chain tilt angle φ .

Supplementary Fig. 14 | a,c,e, Histograms of φ for the lamellae shown in Supplementary Figs. 11, 12, and 13, respectively. **b,d,f**, Histograms of φ for isolated and stacked lamellae shown in Supplementary Figs. 11, 12, and 13, respectively. Because stacked lamellae were more prevalent in the present specimen, the integrated histograms shown in a, c, and e exhibited similar features to the histograms for the stacked lamellae.

The real problem arises with the reasoning proposed for the selection of small tilt angles in stacked lamellae, which appears to be flawed. Referring back to the original arguments of Sir Charles Frank (Faraday Discuss. Chem. Soc. 1979, 68, 7), which are cited in ref 14, in the absence of significant contributions from chain ends, $(1-p)$ and $\cos(\theta)$ should be inversely related, where p is the fraction of folds and θ is the tilt angle. As tilt angle increases (i.e. $\cos(\theta)$ decreases), the fraction of folds

decreases (i.e. $(1-p)$ increases). That is, low fold density is mitigated by large tilt angle, in contrast to the argument presented by the authors on p15. The reasoning that tie chains form at the expense of folds in stacked lamellae then is not consistent with the observation of small tilt angles in stacked lamellae. Since the methods reported here do not image molecular features like folds or tie chains, the proposed formation mechanism cannot be readily confirmed or refuted. Publication of the manuscript with this newly introduced shortcoming is not recommended.

→ We greatly appreciate the reviewer's suggestion. As the reviewer pointed out, the equation proposed by Sir Charles Frank (Eqs. (R1) and (R2)) indicates that the fraction of sharp folds (p) decreases as the chain tilt angle (θ) increases, as shown in Fig. R1.

$$(1-p) \cos\theta \leq 3/10 \quad (R1)$$

$$\therefore p \geq 1 - 3/(10 \cos\theta) \quad (R2)$$

Fig. R1. Fraction of sharp folds (p) with respect to the chain tilt angle (θ). $p = 1-3/(10\cos\theta)$, assuming an equal sign in Eq. (R2). The values calculated using the equation of Frank and those provided by the simulations of Gautam et al. are indicated by filled and open circles, respectively.

Although the density anomaly derived from loose loops and dangling chains (Fig. R2) is considered in the above equations, the equation of Sir Frank does not consider the volume occupied by the sharp folds (i.e., density anomaly due to the sharp folds). After the paper of Sir Frank was published, Gautam et al. conducted a molecular simulation that considered the density anomaly due to sharp folds. The simulation revealed a similar trend to Sir Frank's equation (Fig. R1). Thus, we agree with reviewer #3 that our previous discussion about the formation mechanism was inconsistent with this trend.

As the reviewer mentioned, our experimental method cannot image molecular chains, e.g. folds and tie chains; hence, the formation mechanism in the previous manuscript was highly speculative. Therefore, we deleted this discussion in the revised manuscript to avoid confusing readers and shortened the corresponding part. Please see below.

Fig. R2. Schematic of chain overcrowding near the surface of the lamellar crystal.

Pages 15, Lines 243–257

Thus, the fraction of sharp folds near the flat faces of the lamellae was presumably the only factor affecting the chain tilt in the lamellae. Theory⁴⁸ and simulations¹³ predicted that the sharp-fold fraction decreases with an increase in the tilt angle. Fritzsche et al. used the sharp-fold fraction of $\sim 1/3$ and the thermodynamically stable φ of 34° to moderate the density anomaly near lamellar surfaces¹⁴. Our experimental result, i.e., smaller φ values in stacked lamellae than in isolated ones, implies that the sharp-fold fractions may differ between stacked and isolated lamellae. This difference may be due to the existence of tie chains that bridge two or more neighbouring lamellae in the case of stacked lamellae⁴⁷.

Nanoscale analysis revealed that two types of lamellar crystals (isolated and stacked) were present in a single specimen and that they exhibited different chain tilt angles (Fig. 4i). The reason for such different tilt angles remains an open question and is expected to be revealed by simulating the crystallisation dynamics near the lamellar surface with the knowledge of the directly measured tilt angles of all the lamellae examined in the present study.

Reviewers' Comments:

Reviewer #3:

Remarks to the Author:

The authors have refined their analysis of tilt angles to demonstrate more clearly the existence of two populations. They have also removed the argument that stacked lamellae have fewer folds, but retained the intuition that such lamellae have more tie chains. No further explanation is attempted. The primary contribution of this work remains the remarkable resolution of the NDI technique, which offers some insight into the complex nanoscale structure of a semicrystalline polymer.

Reply to Reviewer #3

The authors have refined their analysis of tilt angles to demonstrate more clearly the existence of two populations. They have also removed the argument that stacked lamellae have fewer folds, but retained the intuition that such lamellae have more tie chains. No further explanation is attempted. The primary contribution of this work remains the remarkable resolution of the NDI technique, which offers some insight into the complex nanoscale structure of a semicrystalline polymer.

→ We greatly appreciate reviewer #3's careful reading and suggestive comments, which were very helpful in improving our manuscript. As for the reviewer's concern about the tie chains between lamellae, we agree that our discussion is intuitive. Thus, we have removed the corresponding sentences. We made changes to the manuscript as follows, which are highlighted in red.

Pages 15, Lines 250–252

~~This difference may be due to the existence of tie chains that bridge two or more neighbouring lamellae in the case of stacked lamellae⁴⁷.~~

Pages 16, Lines 267–269

While isolated lamellae exhibited φ values close to the thermodynamically favoured angle (34°), stacked lamellae predominantly had φ values smaller than 34° , ~~perhaps owing to the tie chains between neighbouring lamellae.~~